# Genetic associations with parental investment from conception to wealth inheritance in six cohorts

Jasmin Wertz [1] ✉, Terrie E. Moffitt[2,3,4,5,6], Louise Arseneault[3], J. C. Barnes [7], Michel Boivin [8], David L. Corcoran[9], Andrea Danese [3,10,11], Robert J. Hancox [12], HonaLee Harrington[2], Renate M. Houts[2], Stephanie Langevin[2,8], Hexuan Liu[7,13], Richie Poulton [14], Karen Sugden[2,5], Peter T. Tanksley [15,16], Benjamin S. Williams[2,5] & Avshalom Caspi [2,3,4,5,6]

Genetic inheritance is not the only way parents' genes may affect children. It is also possible that parents' genes are associated with investments into children's development. We examined evidence for links between parental genetics and parental investments, from the prenatal period through to adulthood, using data from six population-based cohorts in the UK, US and New Zealand, together totalling 36,566 parents. Our findings revealed associations between parental genetics—summarized in a genome-wide polygenic score—and parental behaviour across development, from smoking in pregnancy, breastfeeding in infancy, parenting in childhood and adolescence, to leaving a wealth inheritance to adult children. Effect sizes tended to be small at any given time point, ranging from RR = 1.12 (95% confidence interval (95%CI) 1.09, 1.15) to RR = 0.76 (95%CI 0.72, 0.80) during the prenatal period and infancy; $\beta$ = 0.07 (95%CI 0.04, 0.11) to $\beta$ = 0.29 (95%CI 0.27, 0.32) in childhood and adolescence, and RR = 1.04 (95%CI 1.01, 1.06) to RR = 1.11 (95%CI 1.07, 1.15) in adulthood. There was evidence for accumulating effects across development, ranging from $\beta$ = 0.15 (95%CI 0.11, 0.18) to $\beta$ = 0.23 (95%CI 0.16, 0.29) depending on cohort. Our findings are consistent with the interpretation that parents pass on advantages to offspring not only via direct genetic transmission or purely environmental paths, but also via genetic associations with parental investment from conception to wealth inheritance.

Parents vary in the extent to which they invest resources, attention, time and money in their children's development. This variation matters because it has the potential to reinforce social inequalities in opportunities. For example, compared to less-educated parents, highly educated parents tend to engage in more developmentally rich activities with their children, such as talking and reading, which have been linked with better educational outcomes[1,2]. These parents are also more likely to be able to provide financial supports to their children as they grow into adulthood, promoting offspring attainment and wealth[3,4]. In many countries, the 'parenting gap' between more and less advantaged parents has been increasing in recent years, mirroring rises in economic inequality[5]. Divergences in parental investment have the potential to constrain equal opportunity and impede social mobility[6,7], making it important to understand what processes contribute to variation

in parental investment and the transfer of resources (both psychological and financial) from parents to children. Here we addressed this question, extending previous research in two ways: first, by incorporating genetic information into analyses of parental investment, and second, by studying parental investment at successive stages throughout the offspring's life course.

It may seem surprising to study associations between genes and parenting, because both are often viewed as separate and even competing forces (that is, 'nature versus nurture'). However, behavioural genetics research shows that genes and many environments—including parenting—are correlated[8–10]. These correlations arise because genes contribute to variation in behaviours that can shape the environments that individuals experience and provide[11,12]. Gene–environment correlations can originate in children's genes; for example, children's genes may be associated with their interest in cognitively stimulating activities and this evokes more stimulation from their parents[13]. Gene–environment correlations can also originate in parents' genes; for example, parents' genes may be associated with the amount of cognitively stimulating activities they provide to their child[14].

Links between parental genes and parenting have four implications. First, they indicate that genes may have environmentally-mediated effects, via parenting. That is, if parental genes are associated with parenting in ways that affect offspring, then this represents an additional pathway of genetic influence on children's outcomes, on top of genetic transmission from parent to child[15,16]. Second, links between genes and parental behaviours could create an accumulation of advantages within families to the extent that children receive a 'double whammy' of genes and environments associated with particular outcomes. That is, if the advantage of inheriting genes associated with educational success is intertwined with the advantage of having these genes in the parents, it could amplify advantages across generations. Third, these links raise an issue of confounding. That is, if the genes and parenting that parents provide to their children are correlated, it means that associations between parenting and child outcomes could partly be attributable to genes[17], and associations between genes and child outcomes could partly be attributable to parenting. Fourth, links between genes and parental behaviours do not mean that parental behaviours are fixed, but instead point towards potentially modifiable environmental mediators of genetic influences. That is, if parental genes are associated with consequential aspects of parenting, environmental interventions that modify these aspects of parenting have the potential to intervene in the path from genes to child outcomes.

So far, there have been limited empirical tests of associations between parental genes and parenting, because such tests require two pieces of information: information about parent and child genetics, and information about parenting. Quantitative-genetic designs, such as twin samples, are mostly of twins-as-children, rather than twins-as-parents[9,18]. Of the twins-as-parents studies that are available[19], few contain extensive data on parenting across development. Molecular-genetic studies with genome-wide information on parents and children offer new opportunities, but until recently, such data were not available in cohorts that also contain detailed assessments of parenting. Here we conducted an investigation of links between parents' genes and parenting, using data from six population-based cohorts in the UK, US and New Zealand, some of which have only recently added genetic data (Table 1). Together, the sample size totals over 30,000 parents. To measure parents' genetics, we constructed genome-wide aggregate measures, polygenic scores, that capture genetic associations with educational attainment, based on a recent genome-wide association study of this phenotype[20]. We focused on genetic variants associated with educational attainment because parental educational attainment is a key dimension along which parental investments vary[2,3].

To select measures of parental investment, we adopted a life-course perspective on parental investment. Parenting and its effects are typically measured during one developmental period,

**Table 1 | Summary of study cohorts**

| Cohort | *n* of parents | Country | Developmental period of child's life covered by cohort assessments | Genotyped family members included in analyses |
|---|---|---|---|---|
| ALSPAC | 7,588 | UK | prenatal, infancy, childhood, adolescence | |
| MCS | 10,313 | UK | prenatal, infancy, childhood, adolescence | |
| E-Risk | 880 | UK | prenatal, infancy, childhood, adolescence | |
| Dunedin | 654 | NZ | childhood | |
| HRS | 8,652 | USA | adulthood | |
| WLS | 8,479 | USA | adulthood | |

*n* refers to the number of parents of European-descent within each cohort who had polygenic-score data. For ALSPAC and E-Risk, the *n* refers to mothers, for MCS, Dunedin, HRS and WLS, the *n* refers to mothers and fathers.

most often childhood. However, parental investment begins from before a child is born (for example, in the form of parental behaviours during pregnancy) and continues after children leave the family home (for example, in the form of continued financial support and wealth inheritances). Children continue to receive a variety of types of support from their parents well into adulthood[21] and wealth inheritances from parents to children may contribute to the intergenerational persistence of wealth[22]. A life-course perspective considers changing forms of parental investment across development and captures the cumulative advantage that can arise if greater parental investments in one developmental period are correlated with investments in another[23].

A model depicting the paths we test in our analyses is presented in the top half of Fig. 1. Path b is the path that we focus on in our study; it depicts the possibility that parents' genes are associated with the parenting they provide to their children. To test this possibility, it is necessary to control for path a, which depicts genetic transmission from (biological) parent to child, and path c, which depicts the possibility that children's genes are also associated with the parenting they receive (this is often referred to as evocative gene–environment correlation or child effects). If paths a and c are not controlled for, associations between parental genes and parenting (path b) may reflect genetic transmission (path a) and evocative gene–environment correlations (path c). We therefore controlled for children's polygenic score in our models. Note that our study did not test offspring developmental

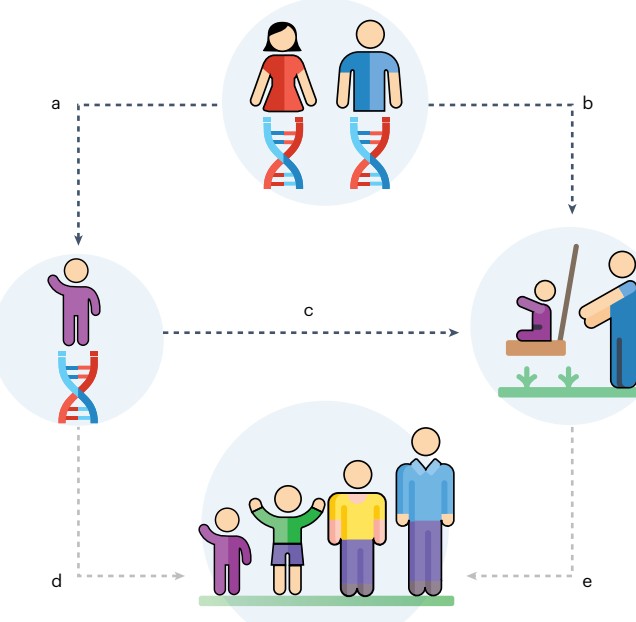

**Fig. 1 | A model of the associations tested or assumed in the present study.** The paths tested (dark dotted lines) or assumed (light dotted lines) in the present study. Path b is the path focused on in this study; it depicts the possibility that parents' genes are associated with the parenting they provide to their children. To test this possibility, it is necessary to control for path a, which depicts genetic transmission from (biological) parent to child, and path c, which depicts the possibility that children's genes are also associated with the parenting they receive (this is often referred to as evocative gene–environment correlation or child effects). If paths a and c are not controlled for, associations between parental genes and parenting may reflect genetic transmission and evocative gene–environment correlations (paths a*c). We therefore controlled for children's polygenic score in the models. Note that this study did not test offspring developmental outcomes (such as attainment or health outcomes), which are represented at the bottom of the figure and which are connected to the top of the figure by paths d and e. It is assumed, on the basis of previous literature (Supplementary Table 1), that the genes parents pass on and the parenting they provide both affect offspring outcomes (in the figure, this is illustrated by paths a*d for genes, and paths b*e for parenting). Also note that even though the parent icon shows both mothers and fathers; most of the analyses used maternal polygenic score due to data availability; fathers' polygenic scores were analysed in only a subset of models.

outcomes (such as attainment or health outcomes), which are represented at the bottom of Fig. 1 and which are connected to the top of the figure by paths d and e. It is assumed, on the basis of previous literature (summarized in Supplementary Table 1) that the genes parents pass on and the parenting they provide both affect offspring outcomes (in the figure, this is illustrated by paths a*d for genes, and paths b*e for parenting).

We tested the hypothesis that parents' education polygenic scores would be associated with life-course parental investments from the prenatal period through to offspring infancy, childhood, adolescence and into adulthood. For each developmental period, we selected aspects of parental investment that have been the focus of previous research, and that are thought to have an impact on offspring health, wealth and wellbeing (Table 2 and Supplementary Table 1). To test whether associations were due to evocative child effects on parenting, we re-ran models controlling for children's polygenic scores. To test how associations between parental genetics and parenting are intertwined with measures of parental socioeconomic advantage, we analysed parents' educational attainment. To test more specifically the role of fathers (in addition to mothers), we conducted a separate set of analyses of them.

**Table 2 | Overview of measures of parental investment assessed in each cohort**

| | ALSPAC | E-Risk | MCS | Dunedin | HRS | WLS |
|---|---|---|---|---|---|---|
| **Prenatal (pregnancy)** | | | | | | |
| Smoking | x | x | x | | | |
| Heavy drinking | x | | x | | | |
| **Infancy (0–1 years)** | | | | | | |
| Breastfeeding | x | x | x | | | |
| **Childhood (2–11 years)** | | | | | | |
| Cognitive stimulation | x | x | x | x | | |
| Warmth, sensitivity | x | x | x | x | | |
| Low household chaos | x | x | x | | | |
| Health-parenting | x | x | x | | | |
| School support | x | | x | | | |
| **Adolescence (12–18 years)** | | | | | | |
| Monitoring | x | x | x | | | |
| **Adulthood (19+ years)** | | | | | | |
| Financial support | | | | | x | x |
| Childcare support | | | | | x | x |
| Inheritance | | | | | x | x |

## Results

### Genetic associations with health habits during pregnancy

We analysed genetic associations with health habits during pregnancy—cigarette smoking and heavy alcohol drinking—in the ALSPAC (Avon Longitudinal Study of Parents and Children), MCS (Millennium Cohort Study) and, for prenatal smoking only, E-Risk (Environmental Risk Longitudinal Twin Study) cohorts. We constructed binary measures to indicate smoking and heavy drinking on the basis of mothers' reports. In each cohort, mothers with higher education polygenic scores were less likely to smoke during pregnancy (ALSPAC (relative risk) RR = 0.76 (95% confidence interval (95%CI) 0.72, 0.80), $P < 0.001$; E-Risk RR = 0.85 (95%CI 0.75, 0.97), $P = 0.014$; MCS RR = 0.76 (95%CI 0.72, 0.80), $P < 0.001$; Fig. 2a). In the ALSPAC cohort but not in MCS, mothers with higher education polygenic scores were also less likely to drink heavily during pregnancy (ALSPAC RR = 0.87 (95%CI 0.82, 0.91), $P < 0.001$; MCS RR = 0.97 (95%CI 0.84, 1.13), $P = 0.712$; Fig. 2b).

### Genetic associations with breastfeeding during infancy

We analysed genetic associations with breastfeeding in the ALSPAC, E-Risk and MCS cohorts. We constructed a binary measure to indicate whether mothers reported ever breastfeeding their children. In each cohort, mothers with higher education polygenic scores were more likely to have breastfed their children (ALSPAC RR = 1.12 (95%CI 1.09, 1.15), $P < 0.001$; E-Risk RR = 1.24 (95%CI 1.13, 1.37), $P < 0.001$; MCS RR = 1.12 (95%CI 1.10, 1.14), $P < 0.001$; Fig. 2c).

### Genetic associations with parenting during childhood

We analysed genetic associations with five aspects of parenting and the home environment: cognitive stimulation (the extent of caregivers' efforts to enrich their child's development); warmth and sensitivity (the extent of caregivers' expressions of affection and responsiveness towards the child); low household chaos (the extent of noise, crowding and set routines in children's homes); health-parenting (the extent of caregivers' attempts to promote healthy habits in children) and school support (the extent of caregivers' involvement in and ambitions for children's schooling). Data came from the ALSPAC, E-Risk, MCS and Dunedin cohorts (although

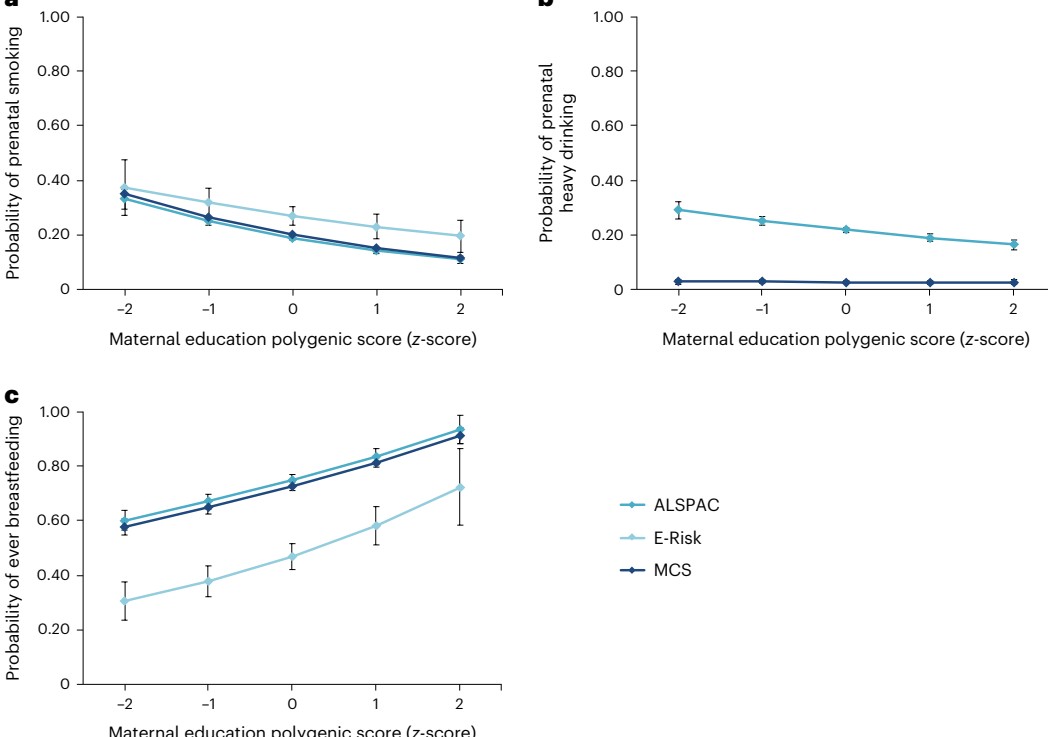

**Fig. 2 | Associations between parental polygenic scores and parenting during pregnancy (prenatal smoking; prenatal heavy drinking) and infancy (breastfeeding). a–c**, The predicted mean probabilities of each outcome across the distribution of *z*-standardized maternal polygenic scores, separately for prenatal smoking (**a**); prenatal heavy drinking (**b**) and breastfeeding (**c**). Not all measures were available in each cohort (for example, measures of heavy prenatal drinking were only available in the ALSPAC and MCS cohorts). Note, the E-Risk sample is a twin sample, hence rates of breastfeeding are lower than those in the ALSPAC and MCS cohorts. All effect sizes are reported for a 1-s.d. change in polygenic score. The error bars indicate 95% confidence intervals. The number of participants (mothers) included in the analysis were as follows: for prenatal smoking ALSPAC *n* = 7,190; E-Risk *n* = 846; MCS *n* = 6,690; for prenatal heavy drinking ALSPAC *n* = 7,144; MCS *n* = 6,695; for breastfeeding ALSPAC *n* = 7,025; E-Risk *n* = 855; MCS *n* = 6,222.

not every cohort contained every measure). Outcomes were assessed using a variety of measures: ratings of parent–child interactions; interviewer observations of the home and parent, child and teacher responses to questionnaires (Table 2 and Supplementary Information). Figure 3 reports the associations between mothers' polygenic scores and parenting. In each cohort, mothers with higher education polygenic scores tended to provide more cognitive stimulation and warm, sensitive parenting; to raise their children in less chaotic households; to promote healthier habits and to provide more school support (Fig. 3).

Genetic associations with caregiving may arise due to children's rather than mothers' genetics, if children's genetic differences evoke differences in parental behaviour (that is, 'child effects' on parenting). For example, children with higher education polygenic scores may evoke more cognitive stimulation from their mothers via child behaviours such as earlier talking or better reading skill[13,24]. To account for this possibility, we incorporated children's education polygenic scores into our models (Fig. 4). Children's polygenic scores were associated with most parenting measures (Supplementary Fig. 1). Figure 4 reports the results of adding children's polygenic scores to a model containing mothers' polygenic scores, in those cohorts where maternal and child genetic data were available. Adding children's polygenic scores reduced associations between mothers' polygenic scores and parenting by approximately 40%, on average, with some variation across cohorts (Fig. 4). However, as Fig. 4 shows, mothers' polygenic scores remained associated with most measures of parenting, suggesting that regardless of child genetics, mothers with higher education polygenic scores tend to provide greater parental investment.

## Genetic associations with monitoring during adolescence

We analysed genetic associations with parental monitoring, that is, the extent of parents' knowledge and rule-setting about their children's activities and whereabouts, in the ALSPAC, E-Risk and MCS cohorts. Parental monitoring was assessed through maternal and child reports. Results are presented in Fig. 3 and show that, in the E-Risk and MCS cohorts, mothers with higher education polygenic scores tended to monitor their children more closely (Fig. 3). However, after adjusting for children's polygenic scores, maternal genetic associations with monitoring were small and statistically non-significant, suggesting that genetic associations with monitoring can best be explained by child effects. These analyses are reported in Fig. 4 and Supplementary Fig. 1. This finding is consistent with previous research reporting child effects on parental monitoring[25,26].

## Genetic associations with supports to adult children

We analysed genetic associations with three indicators of intergenerational supports from parents to adult children: financial support; help with childcare and intention of leaving a wealth inheritance. Data came from the HRS (Health and Retirement Study) and WLS (Wisconsin Longitudinal Study) cohorts. Parents with higher education polygenic scores were more likely to provide financial support to their children (HRS RR = 1.12 (95%CI 1.10, 1.14), *P* < 0.01; WLS RR = 1.06 (95%CI 1.04; 1.09; *P* < 0.001) and help with childcare (HRS RR = 1.03 (95%CI 1.01, 1.06), *P* < 0.002; WLS RR = 1.10 (95%CI 1.05, 1.14), *P* < 0.001). In HRS only, parents' polygenic scores were also associated with the intention of leaving a wealth inheritance (HRS *β* = 0.12 (95%CI 0.11; 0.14), *P* < 0.001; WLS RR = 1.00 (95%CI 0.98, 1.02), *P* = 0.582). Supplementary Table 3

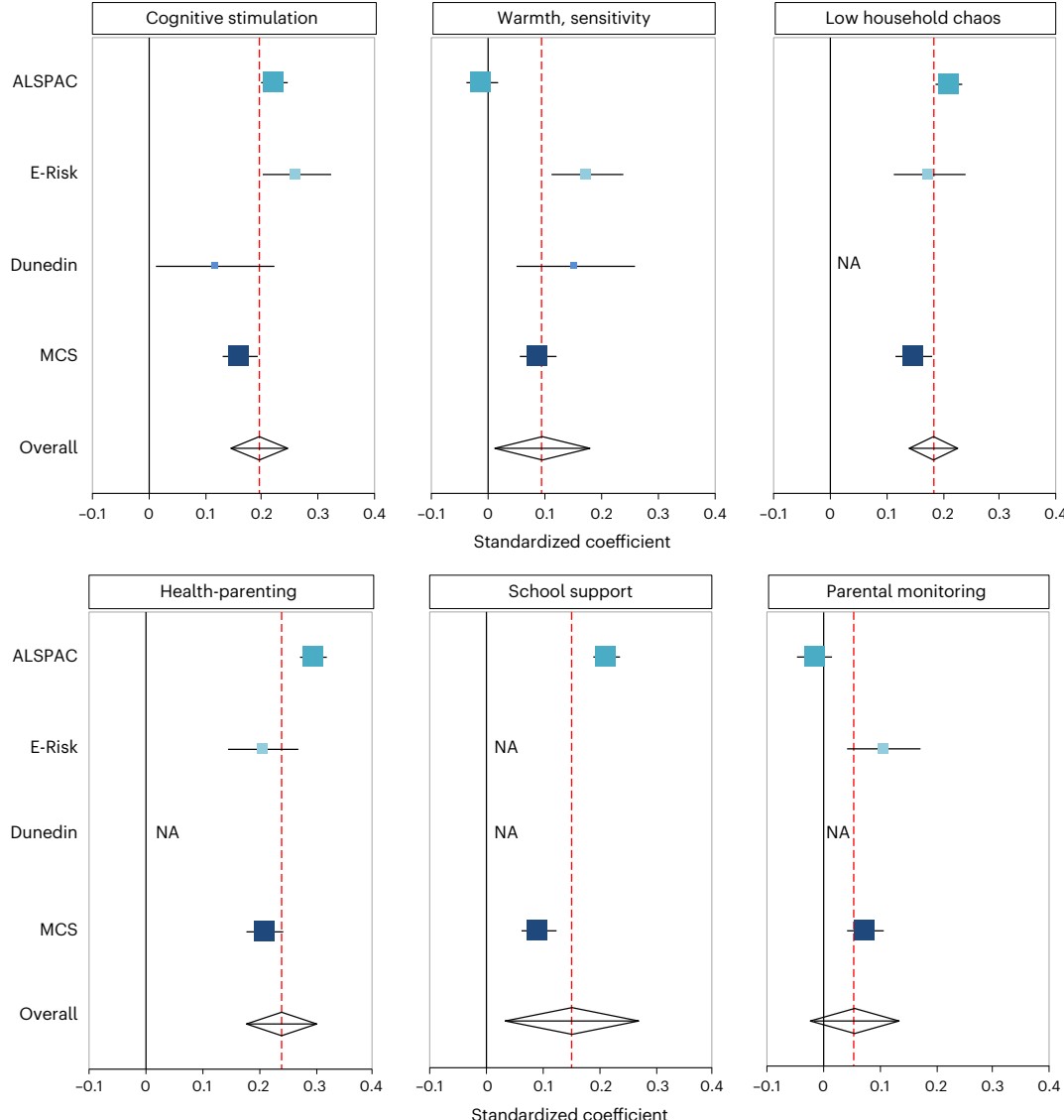

**Fig. 3 | Associations between maternal polygenic scores and parenting during childhood and adolescence.** Associations (expressed as standardized regression coefficients) between maternal education polygenic scores and measures of parenting during childhood (cognitive stimulation; warmth, sensitivity; low household chaos; health-parenting; school support) and adolescence (parental monitoring) in the ALSPAC, E-Risk, Dunedin and MCS cohorts. Not all measures were available in each cohort (for example, measures of school support were only available in the ALSPAC and MCS cohorts). The overall effect was calculated using a random-effects model. The centre of the effect marker indicates the estimate of the association between polygenic score and parenting, expressed as a standardized regression coefficient. The error bars indicate 95% confidence intervals. The size of the effect size markers corresponds to the sample size, so that larger sample sizes have larger markers. The number of participants (mothers) included in the analysis were as follows: for cognitive simulation ALSPAC $n = 6,180$; E-Risk $n = 879$; Dunedin $n = 333$; MCS $n = 5,238$; for warmth, sensitivity ALSPAC $n = 5,226$; E-Risk $n = 880$; Dunedin $n = 330$; MCS $n = 5,382$; for low household chaos ALSPAC $n = 6,210$; E-Risk $n = 878$; MCS $n = 5,268$; for health-parenting ALSPAC $n = 5,649$; E-Risk $n = 877$; MCS $n = 5,268$; for school support ALSPAC $n = 6,603$; MCS $n = 5,385$; for parental monitoring ALSPAC $n = 4,092$; E-Risk $n = 866$; MCS $n = 5,580$. NA, not applicable.

shows that results did not substantively change when adjusting for respondents' age, sex, study year/wave, number of children, proximity (for childcare), labour force status and assets/net worth.

**Genetic associations with cumulative parental investment**
So far, our analyses show genetic associations with parenting for one developmental period at a time. However, this approach might not capture the full magnitude of associations with parenting, if greater parental investment in one period is associated with greater investment in another. To capture this process, we constructed a measure of the accumulation of parental investment across developmental periods. We conducted these analyses in the E-Risk, MCS and ALSPAC cohorts because these had repeated measures of parenting behaviour across

developmental periods. We created binary variables for each developmental period, capturing 'higher' (versus 'lower') parental investment, based on not smoking/drinking during pregnancy, breastfeeding in infancy and scoring in the upper 50% for all parenting measures during childhood and adolescence. In all three cohorts, children who received greater parental investment in one developmental period tended to do so in other periods as well, although associations were generally higher in E-Risk than in MCS and ALSPAC (Fig. 5). We then constructed a measure indicating the accumulation of parental investment across time. The measure was constructed by adding up the individual indicators of investment for each developmental period, so that the lowest scores indicated consistently low parental investment across development and the highest scores displayed consistently high investment (Fig. 5).

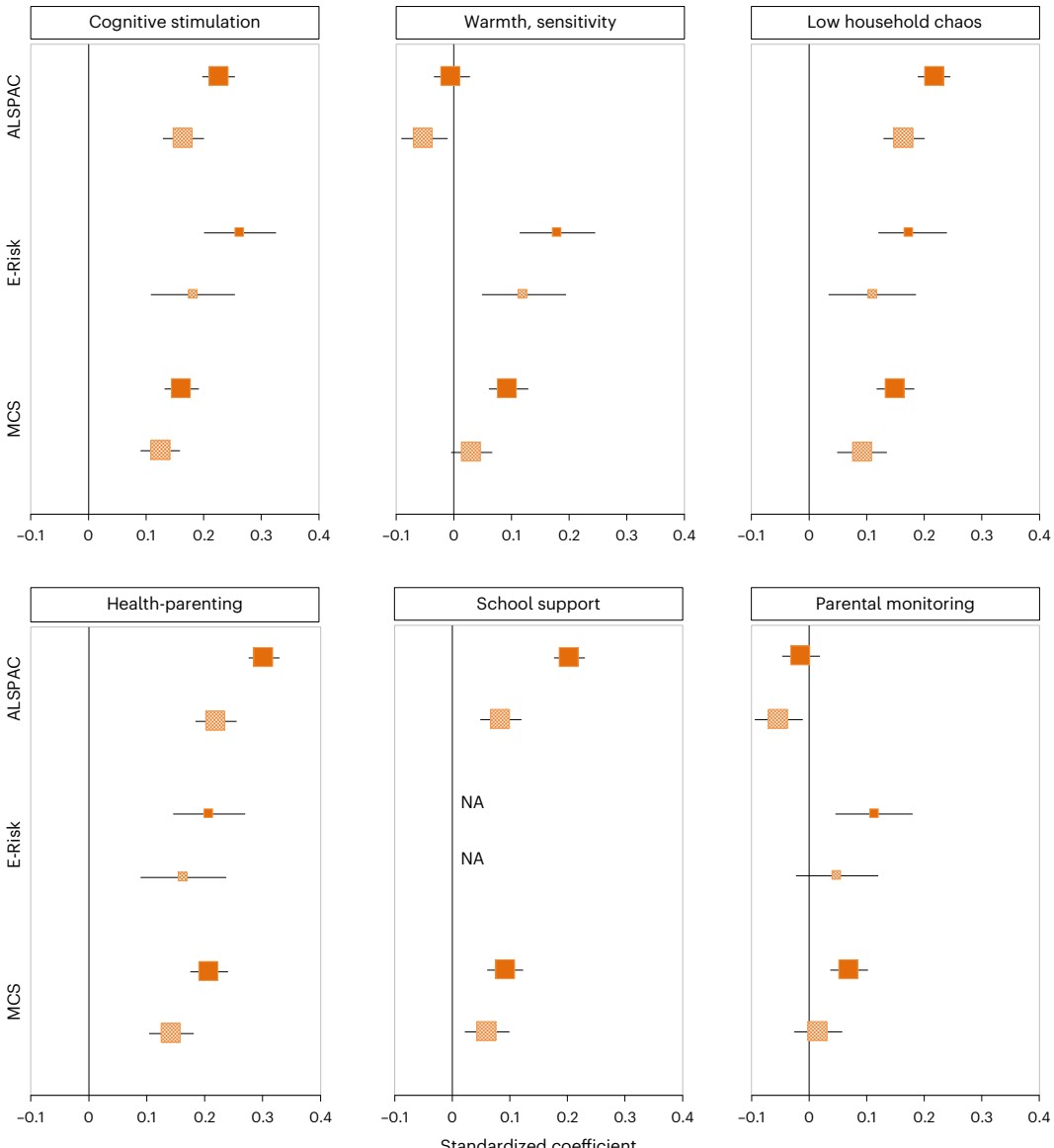

**Fig. 4 | Associations between maternal polygenic scores and childhood parenting adjusted for children's polygenic scores in the ALSPAC, E-Risk and MCS cohorts.** Associations (expressed as standardized regression coefficients) between maternal education polygenic scores and measures of parenting during childhood (cognitive stimulation; warmth, sensitivity; low household chaos; health-parenting; school support) and adolescence (parental monitoring) in the ALSPAC, E-Risk and MCS cohorts before (dark orange boxes) and after (patterned orange boxes) adjusting for children's education polygenic scores (the Dunedin cohort is not included because it does not contain measures of child genetics). Not all measures were available in each cohort (for example, measures of school support were only available in the ALSPAC and MCS cohorts). The centre of the effect marker indicates the estimate of the association between polygenic score and parenting, expressed as a standardized regression coefficient. The error bars indicate 95% confidence intervals. The size of the effect size markers corresponds to the sample size, so that larger sample sizes have larger markers. The number of participants (mother–child dyads) included in the analysis were as follows: for cognitive simulation ALSPAC $n = 4,342$; E-Risk $n = 859$; MCS $n = 5,093$; for warmth, sensitivity ALSPAC $n = 3,926$; E-Risk $n = 860$; MCS $n = 5,225$; for low household chaos ALSPAC $n = 4,451$; E-Risk $n = 858$; MCS $n = 5,117$; for health-parenting ALSPAC $n = 4,093$; E-Risk $n = 858$; MCS $n = 5,124$; for school support ALSPAC $n = 4,586$; MCS $n = 5,228$; for parental monitoring ALSPAC $n = 3,343$; E-Risk $n = 847$; MCS $n = 5,414$.

In all three cohorts, mothers with higher polygenic scores tended to provide consistently greater parental investment across time (E-Risk $\beta = 0.23$ (95%CI 0.16, 0.29), $P < 0.001$; MCS $\beta = 0.21$ (95%CI 0.18, 0.25), $P < 0.001$; ALSPAC $\beta = 0.15$ (95%CI 0.11, 0.18), $P < 0.001$). This association reduced but persisted after controlling for children's polygenic scores (E-Risk $\beta = 0.15$ (95%CI 0.07, 0.22), $P < 0.001$; MCS: $\beta = 0.15$ (95%CI 0.11, 0.19), $P < 0.001$; ALSPAC $\beta = 0.09$ (95%CI 0.04, 0.13), $P < 0.001$). The difference in mean polygenic score among mothers of children who received high parental investment in all versus none of the developmental periods amounted to approximately 0.8 standard deviations (Fig. 5).

**Parental education and genetic associations with parenting**
We next tested how genetic associations with parenting were intertwined with parents' educational attainment. Adjusting for parents' educational attainment reduced genetic associations with parenting by approximately two-thirds (Supplementary Table 4). This finding was consistent across all parenting outcomes at all offspring life stages. The reduction in the size of genetic associations suggests at least two possibilities. First, the polygenic score may tap resources that accrue with additional years of education and that shape parenting practices, including financial resources or greater knowledge about parenting[2].

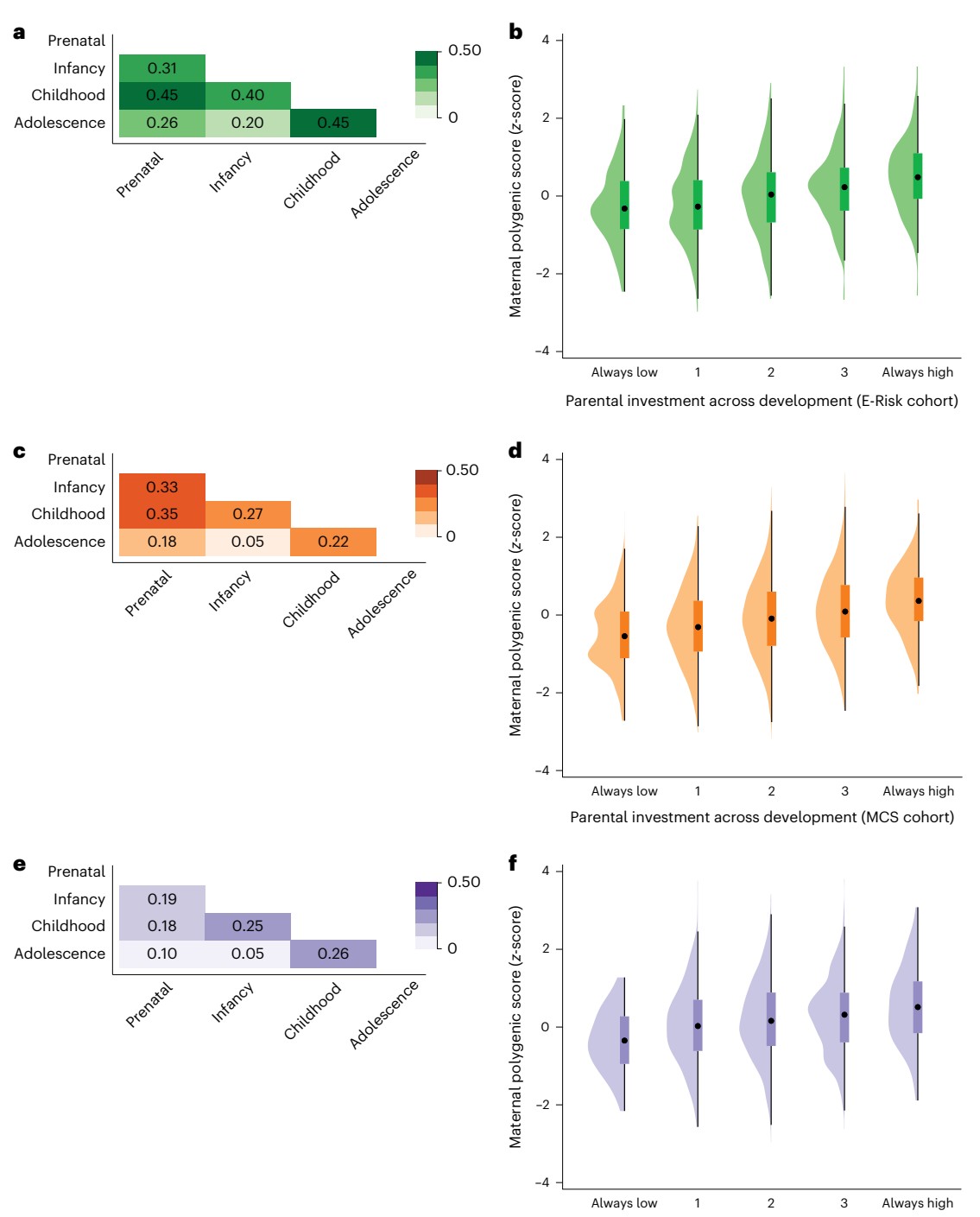

**Fig. 5 | Cumulation of parental investment across development in the E-Risk, MCS and ALSPAC cohorts. a–f**, Analyses from mothers and children participating in the E-Risk (green), MCS (orange) and ALSPAC (purple) cohorts. **a,c,e**, Tetrachoric correlations between parental investment across developmental periods: E-Risk (**a**), MCS (**c**) and ALSPAC (**e**). For each developmental period (prenatal, infancy, childhood, adolescence), a binary variable was constructed, capturing 'high' parental investment (see Results for details). The colours in the matrices indicate strength of association, with darker areas indicating higher correlations. The panels show that high parental investment in one period tended to be associated with high parental investment in other periods. **b,d,f**, The median, interquartile range and distribution of education polygenic scores for each category of a measure indicating the cumulation of parental investment across time: E-Risk (**b**), MCS (**d**) and ALSPAC (**f**). The measure of cumulative parental investment was constructed by adding

up the individual indicators of investment for each developmental period, so that the lowest scores indicated consistently low investment ('Always low', $n = 189$ in E-Risk; $n = 425$ in MCS; $n = 119$ in ALSPAC) and the highest scores indicated consistently high investment ('Always high', $n = 168$ in E-Risk; $n = 255$ in MCS; $n = 194$ in ALSPAC). In between these extreme categories were categories indicating lower investment ($n = 460$ in E-Risk; $n = 1,313$ in MCS; $n = 633$ in ALSPAC), a moderate amount of investment ($n = 490$ in E-Risk; $n = 2,594$ in MCS; $n = 1,422$ in ALSPAC) or higher investment ($n = 349$ in E-Risk; $n = 1,655$ in MCS; 1,123 in ALSPAC). The point in each box boxes indicates the median, the box indicates the interquartile range around the median and the shaded area indicates the distribution of the polygenic score. The number of participants included in the analysis were as follows: for E-Risk $n = 1,656$; for MCS $n = 6,242$ and for ALSPAC $n = 3,491$.

**Table 3 | Polygenic scores of family members depending on father characteristics**

| | Mother polygenic score | | | | Child polygenic score | | | | Father polygenic score | | | |
|---|---|---|---|---|---|---|---|---|---|---|---|---|
| | **M** | **s.d.** | **n** | **P** | **M** | **s.d.** | **n** | **P** | **M** | **s.d.** | **n** | **P** |
| ALSPAC (child age 4) | | | | | | | | | | | | |
| Resident biological father | 0.08 | 1.00 | 3,623 | | 0.07 | 0.99 | 3,623 | | – | – | – | |
| Non-resident biological father | −0.21 | 0.97 | 406 | <0.001 | −0.24 | 1.00 | 406 | <0.001 | – | – | – | |
| MCS (child age 14) | | | | | | | | | | | | |
| Father genotyped | 0.01 | 0.99 | 2,501 | | 0.04 | 0.99 | 2,501 | | – | – | – | |
| Father not genotyped | −0.24 | 1.00 | 2,918 | <0.001 | −0.24 | 0.98 | 2,918 | <0.001 | – | – | – | |
| Dunedin (child age 3) | | | | | | | | | | | | |
| Father participated in Parenting assessment | – | – | – | – | – | – | – | – | 0.02 | 1.00 | 316 | |
| Father did not participate | – | – | – | – | – | – | – | – | −0.22 | 0.98 | 29 | 0.218 |

The numbers reported are means and standard deviations of family members' z-standardized polygenic scores (in MCS and ALSPAC, mother and child polygenic scores were standardized within the sample of mothers and children who had a polygenic score; in the Dunedin cohort, fathers' polygenic scores were standardized within the sample of fathers who had a polygenic score). We compared means using linear regression models; all tests were two-tailed; no adjustments were made for multiple comparisons.

Second, the polygenic score may tap personal characteristics (for example, cognitive skills and self-control) that are associated with differences in educational attainment as well as in parenting[27,28].

### Fathers' genetics and parenting

Most research on parenting focuses on caregiving provided by mothers[29]. Here we additionally explored associations between fathers' polygenic scores and parenting. A challenge with studying fathers is that they can be difficult to include in studies of child development, because not all fathers live with their children. As a result, fathers who participate in research (and with their families) may not be representative of the whole population of fathers (and families)[30]. We tested this selection effect at a genetic level in three different ways. First, using ALSPAC data, we tested whether mothers' and children's polygenic scores differed by whether the child's biological father resided with the family. Second, using MCS data, we tested whether mothers' and children's polygenic scores differed by whether the child's family included a genotyped father or not. Third, using Dunedin data, we exploited that genotyping had been done independently of fathers' participation in the substudy that included parenting assessments. Findings show that if fathers resided with their families (ALSPAC), were genotyped (MCS) and participated in parenting assessments (Dunedin), family members' polygenic scores tended to be higher (Table 3), indicating selection effects at a genetic level. Studies that include fathers, including genetic studies of mother, father and child 'trios', risk representing a select subset of individuals across both generations.

With these caveats in mind, we tested associations between fathers' polygenic scores and parenting within the selected subsamples of fathers, in two ways. First, using data from the Dunedin and MCS cohorts, we analysed associations with fathers' cognitively stimulating and warm-sensitive parenting, as assessed using video-taped recordings of fathers interacting with their children (in the Dunedin cohort) and father's self-report questionnaires (in the MCS cohort). In both cohorts, fathers' polygenic scores were associated with cognitive stimulation (Dunedin $\beta = 0.13$ (95%CI 0.02; 0.24), $P = 0.023$; MCS $\beta = 0.15$ (95%CI 0.10; 0.19), $P < 0.001$) with similar effect sizes to those for mothers using comparable measures (that is, Dunedin $\beta = 0.12$ (95%CI 0.01; 0.22), $P = 0.028$; MCS $\beta = 0.16$ (95%CI 0.13; 0.19), $P < 0.001$). In the Dunedin cohort only, fathers' polygenic scores were also associated with warm, sensitive parenting (Dunedin $\beta = 0.12$ (95%CI 0.01; 0.23), $P = 0.032$; MCS $\beta = 0.03$ (95%CI −0.01; 0.07), $P = 0.142$); these results were similar to those for mothers using comparable measures (that is, Dunedin $\beta = 0.15$ (95%CI 0.05; 0.26), $P = 0.004$; MCS $\beta = 0.06$

(95%CI 0.03; 0.09), $P < 0.001$). Thus, within the subset of fathers for whom data were available, genetic associations with parenting tended to be similar for fathers and mothers.

Second, using data from the MCS cohort, where genetic data for mothers, fathers and children were available in a subset of families ($n = 2,503$), we added fathers' polygenic scores to models containing mothers' and children's polygenic scores. These results are reported in Supplementary Fig. 2. They show that fathers' polygenic scores were uniquely associated with several parenting outcomes, over and above mothers' and children's polygenic scores, specifically cognitive stimulation; warm, sensitive parenting and health-parenting. It is also notable that in these models, children's polygenic scores were no longer uniquely associated with most parenting outcomes (except parental monitoring), suggesting that much of the apparent child effect on parenting in models that do not adjust for fathers may reflect fathers' genetics, at least in this subset of genotyped trios. Furthermore, although there was evidence of assortative mating in the MCS cohort (mothers' and fathers' polygenic scores were correlated $r = 0.14$ (95%CI 0.09; 0.19), $P < 0.001$), the results reported in Supplementary Fig. 2 show that mothers' polygenic scores remained associated with most parenting outcomes after adjusting for fathers' polygenic scores.

### Discussion

Families play a major role in structuring children's access to resources and opportunities from birth through adulthood. Examining the processes that are associated with parents' investment in their children may contribute to a better understanding of the intergenerational transmission of social inequalities. Here, we added to previous research by bringing together genetic data of parents and children with rich assessments of parental behaviour to test how genes and parental investment combine across the life course. The findings reveal widespread associations between parental education-associated genetics and parental behaviour across development. Across cohorts, children of mothers with higher education polygenic scores tended to experience a healthier in utero environment; were more likely to be breast-fed; grew up to receive more cognitively stimulating, health-oriented, structured parenting and support with their schooling and, as adults, were more likely to receive time and monetary supports from their parents. For most outcomes, parents' genetics were associated with parenting net of children's genetics, indicating that children were exposed to advantageous environments on top of genetic transmission from parent to child. Individual associations tended to be small[31] in each developmental period, but there was evidence for accumulating

effects across development. Our findings are consistent with the interpretation that parents bestow advantages on offspring not only via direct genetic transmission or purely environmental paths, but also via genetic associations with parental investments from conception to leaving a wealth inheritance.

Our findings should be interpreted in light of limitations. First, we did our best to harmonize measures across cohorts, but this was not always possible. For example, intentions of leaving an inheritance were measured using different formats across cohorts, and this may partly explain variation in results for this outcome. Furthermore, some aspects of parenting had better measurement coverage than others (for example, across cohorts, warm, sensitive parenting and household chaos tended to have fewer measurement occasions than cognitive stimulation). However, for the most part results were consistent across cohorts, suggesting that the findings did not depend on one particular approach to measuring parenting. Second, although we were able to control for genetic child effects when testing associations with parenting up to adolescence, we could not control for genetic child effects on parental investment in adult children. To our knowledge, there are no studies that have these data. However, if effects of adult offspring on parenting are similar to those we observed earlier in childhood, we would expect a reduction of these associations by about 40%. Third, like much of behavioural genetic research, our sample only includes individuals of European ancestry[32]. More research in diverse samples is needed. Our findings do not provide any information whatsoever about origins or correlates of any racial differences in educational attainment or parenting. Fourth, some dimensions of parental investment—such as the amount of time spent with children or children's engagement in extracurricular activities—were not included because they had not been assessed consistently across the studies we analysed. Fifth, sample dropout over time was substantial in some of the cohorts (for example, approximately 50% dropout in the ALSPAC cohort by adolescence), which risks introducing bias[33]. Associations replicated in the E-Risk and Dunedin cohorts, which have had very high retention over the years (more than 90%). Sixth, the number of fathers with genotype data was smaller than anticipated, and selection effects were evident. There were also not as many parenting measures available and reported by fathers as there were for mothers. The lack of data from fathers is a well-described issue in developmental science[29]. This means that associations between children's polygenic score and parenting net of mothers' polygenic scores—which are often interpreted as an evocative effect—could partly reflect fathers' genetics. This is corroborated by our analyses in MCS, which showed null associations between children's polygenic scores and most parenting outcomes after adjusting for fathers' (and mothers') polygenic scores. In addition to this issue, to the extent that mothers' and fathers' genes are correlated, for example because of assortative mating[34], associations between mothers' genetics and parenting could partly reflect fathers' genetics. However, even after adjusting for fathers' polygenic scores in MCS, mothers' polygenic scores continued to be associated with most parenting outcomes. Seventh, although the polygenic score used in our study accounts for a considerable portion of variance in educational attainment (approximately 10%) (ref. [20]) this is slightly lower than the estimated single-nucleotide polymorphism (SNP) heritability (approximately 15%) (ref. [20]) and far lower than the estimated twin heritability (approximately 40%)[35]. Our effect sizes may change with future iterations of the polygenic score that capture more heritability. Eighth, childhood periods were covered by NZ and UK cohorts, whereas adulthood was covered by US cohorts, which is a potential source of bias. We could not locate appropriate non-US studies covering the adulthood period. Finally, genetic associations with parenting behaviour in any given developmental period were mostly small. This is not surprising, given that parenting is a complex and dynamic behaviour that is multi-determined by characteristics of the parent, child, partnership, social network and broader cultural, community, societal

and historical context[5,36,37]. Although effects were small, they were consistent across development, and they replicated across cohorts. Furthermore, associations across successive developmental periods had accumulating and larger effect sizes[38].

Our work extends three lines of previous research. First, biometric studies in twins and adoptees have shown broad genetic influences on parenting behaviour[9]. Our results add to these findings, indicating that a narrower genetic signature identified in genome-wide association studies (GWAS) of educational attainment is associated with parental investments across the life course. Together, these findings provide evidence that parenting is partly heritable, which is sometimes referred to as the 'nature of nurture'[39]. Alongside environmental influences, genetic influences may partly explain why parenting tends to 'runs in families', that is, why there tends to be resemblance in parenting behaviours across siblings and across generations[40]. Second, initial studies that combined parental polygenic scores and parental behaviour reported genetic associations with measures of parenting behaviour during children's early years[41,42]. Here we observed that these links are not unique to a few aspects of parenting in the early years of life, but that genetic associations are widespread across measures of parental investment, and continue into offspring adulthood. Third, molecular-genetic studies of parent–offspring trios show that genes that are not passed on from parents to offspring still affect offspring outcomes[15,16,43]. This finding suggests that genes are associated with parental characteristics in ways that affect offspring development (referred to as 'genetic nurture')[34]. Although our study does not examine child outcomes, our analyses point to specific parental behaviours across development that could mediate these effects.

Findings of genetic nurture have typically been interpreted as indicating an effect of parenting (or other parental characteristics) on offspring outcomes, but emerging research suggests a more nuanced interpretation. One recent study by Nivard et al.[44] found that parents' education-associated genetics no longer predicted offspring educational outcomes over and above children's genetics once parents' siblings' genetics were accounted for. This suggests that the processes mediating indirect genetic effects are not specific to one nuclear family, but shared across extended families. A challenge in comparing this and our study is that there are important differences in data and design: the study by Nivard et al.[44] does not analyse parenting directly, whereas ours does; Nivard et al. uses a sibling design, whereas ours does not (because of a lack of twins-as-parents data); Nivard et al. use data from Norway, whereas our study uses data from the UK, New Zealand and the US; Nivard et al. specifically focuses on genetic nurture in relation to offspring educational attainment, whereas we analyse genetic associations with parenting behaviours that have been linked to a broad range of offspring outcomes. Notwithstanding these differences, how do the findings compare? Our findings show some consistency with Nivard et al.[44], because we observe genetic associations with parenting. Genetic associations with parenting imply that parenting will to some extent be shared across genetically related family members. For example, because siblings share genes with each other, and genes are associated with parenting, siblings would naturally be expected to resemble each other in how they parent their offspring, more so than non-siblings. Likewise, there will be some expected parent–child similarity in parenting partly due to genes shared between parents and children. This suggests that parenting and parental investment are not exclusively within-family processes but are to some extent shared across related family members (including siblings and grandparents). This would lead to a reduction of genetic nurture within families, as seen in the study by Nivard et al.[44] without necessarily negating a role for parenting in genetic nurture. However, if genetic nurture would be entirely attributable to parent behaviour, as it has been typically interpreted, one would expect to see at least some residual effect even after introducing a sibling control. Thus, the finding that the genetic nurture effect reduces to zero in the study by Nivard et al.[44], if

replicated, will raise important questions about the interpretation of genetic nurture, including to what extent it reflects processes such as population stratification. Neither the Nivard study nor our study can conclusively answer these questions. Addressing them will require an expansion of extended family studies (such as twins-as-parents studies) that collect molecular-genetic data alongside data on parenting and offspring outcomes.

There are several possible explanations for associations between parental polygenic scores and parenting, some of which we were able to test. A modest portion of the association was accounted for by children's genetics, suggesting that children with higher education polygenic scores evoked greater investment from their parents (that is, evocative gene–environment correlation). This finding is consistent with theory and research on how children shape their own environment, including the parenting they receive[12,25]. However, associations with parental investment mostly persisted net of children's genetics, suggesting that over and above children's genetic variation, parents with higher education polygenic scores tend to differ in their parenting. We also studied the role of parental educational attainment. All associations reduced substantially when parent education was taken into account, although a few associations remained statistically significant. The finding that associations were reduced suggests that associations between parental polygenic scores and parenting are largely due to factors linked with parent education. This could be because parental investment requires resources (for example, money to buy books or healthy foods, or to leave as an inheritance) and knowledge (for example, how to support schooling) that parents with higher polygenic scores may have been more likely to acquire through their education[2]. Alternatively, parents with higher education polygenic scores may differ in personal characteristics (for example, cognitive and self-control skills) that predict both educational attainment as well as parental investment[24]. The finding that some associations remained statistically significant indicates that even net of their education, parents who differ in polygenic scores show some differences in parenting (though residual effects were very small). This suggests that the education polygenic score captures personal characteristics that are associated with individuals' parenting over and above their education[24]. However, it could also reflect sampling or measurement differences between the original GWAS and our cohorts. Another possible explanation for why the education polygenic score is associated with parenting is through its genetic and phenotypic correlations with outcomes other than education, including fertility-related outcomes (such as age at first birth or number of children)[45] or mental health outcomes (for example, depression)[46], which are all associated with parenting. Indeed, previous research in one of our cohorts, the Dunedin Study, shows that the education polygenic score was associated with age at first birth, and that age at first birth was associated with parenting behaviour[14]. However, the study also showed that age at first birth did not explain associations between education polygenic scores and parenting behaviour[14]. Consistent with this finding, in the current study, differences in parental investment observed in WLS and HRS were not explained by number of children. However, more research is needed to systematically compare associations between different polygenic scores and parenting.

The association between education-associated genetics and parenting may have been anticipated on the basis of previous research findings of phenotypic associations between education and parenting[2]. However there are several reasons for also testing these associations using genetic data. For one, a phenotypic association between education and parenting does not in itself mean that there must also be a genetic association between the two, especially given the fact that the polygenic score only explains a fraction of the heritability of educational attainment. Furthermore, the implications of a gene–environment correlation go beyond those of a phenotypic correlation between

education and parenting, because a gene–environment correlation indicates that (1) genes may have environmentally mediated effects through parenting; (2) the unique contributions of genes and parenting cannot be easily separated in research that does not take both into account and (3) the advantage of inheriting genes linked to educational attainment genes is associated with the advantage of invested parenting, resulting in a 'double whammy' of education-associated genes and environments. Although previous studies have reported links between education-associated genes and some aspects of parenting during children's early years, here we provide evidence that these links are not unique to a few aspects of parenting in the early years of life, but are widespread across measures of parental investment, and that they continue and accumulate into offspring adulthood.

Our findings have implications for understanding how genes are associated with the intergenerational transmission of traits and behaviours. Consistent with previous work, our findings suggest that genetic influences may partly operate through environmental conditions that parents create for their children[34]. On the one hand, this indicates that environments could become an extension of genetic tendencies, and thereby potentially reinforce genetic differences between individuals and across generations. Also, environmental conditions created by parents are themselves responsive to interventions, for example through policy changes (for example, tobacco control policies reduce prenatal smoking[47]; wealth inheritance taxation alters financial plans[48]) or direct modification (for example, parent trainings can help change parenting skills[49,50]), offering the potential to intervene in pathways from genes to behaviour.

## Methods

This research complies with all relevant ethical regulations; the board and institutions approving the study protocols are listed in each individual cohort description in the Supplementary Information. Informed consent was obtained from all participants. Details on the compensation of participants is provided in the Supplementary Information for each individual cohort.

### Data sources

We combined data from multiple cohorts to cover parental investment from conception to offspring adulthood (Table 1). Data came from six cohorts: the ALSPAC, the E-Risk Longitudinal Twin Study, the Dunedin Study, the MCS, the WLS and the HRS (Table 1). Together, these cohorts total a sample size of over 30,000 parents. A detailed description of each cohort, including data access procedures, is provided in the Supplementary Information.

### Polygenic scoring

We constructed polygenic scores based on a recent Social Science Genetic Association Consortium (SSGAC) GWAS of educational attainment[20]. In the ALSPAC, E-Risk, Dunedin and MCS cohorts polygenic scores were computed following the method described by Dudbridge[51] using the PRSice software[52]. Briefly, SNPs reported in the GWAS[20] were matched with SNPs in each cohort, regardless of nominal significance for their association with educational attainment[53]. We performed clumping by retaining the SNP with the smallest $P$ value from each linkage disequilibrium block (excluding SNPs with $r^2 > 0.1$ in 500-kb windows), then weighted SNPs by effect estimate, and then summed weighted counts across all genotypes to calculate each participant's polygenic score. In the HRS and WLS cohorts, polygenic scores were computed by the SSGAC using the LDPred software[54]. Because HRS and WLS data were included in the GWAS of educational attainment, polygenic scores for these datasets were constructed using summary statistics after the target dataset was excluded.

Polygenic scoring was restricted to individuals of European ancestry. To account for potential population stratification, polygenic scores

**Article** https://doi.org/10.1038/s41562-023-01618-5

were residualized on the first ten principal components computed from the genome-wide data in each cohort[55]. Residualized polygenic scores were normally distributed and standardized to mean of 0 and s.d. = 1 in each cohort. More details about genotyping and polygenic-score construction are provided in the Supplementary Information.

## Measurement of parenting
Parenting was assessed during five developmental periods of the child's life: prenatal (during pregnancy), infancy (birth to 1 year), childhood (2–11 years), adolescence (12–18 years) and adulthood (19+ years). We selected measures that captured parental investments during each of these periods: health habits in pregnancy; breastfeeding in infancy; parenting in childhood (cognitive stimulation; warm, sensitive parenting; health-parenting; household chaos; school support); parental monitoring in adolescence and support to adult offspring (financial and childcare support; probability of leaving a wealth inheritance). The exact measures differed across cohorts, depending on the data that had been collected in each cohort (Table 2). Measures were harmonized across cohorts as far as possible. Assessment methods included observations of parent–child interactions, informant ratings of the home environment and reports from parents and children (Supplementary Table 3). More detail is provided in the Supplementary Information.

## Data analysis
To analyse binary outcomes, we used Poisson regressions and report relative risks. To present these analyses visually, we used marginsplots as implemented in Stata. Each margins plot reports the predicted probabilities of the outcome at each level of the polygenic score. To analyse continuous outcomes, we used linear regressions and report standardized regression coefficients. To present these analyses visually, we used forest plots. Each forest plot reports an individual estimate for each cohort, and a meta-analysed estimate across cohorts, as obtained using a random-effects model. The distributions of all our continuous parenting outcomes were within a range of −2 to +2 for skewness and −7 to +7 for kurtosis, which are considered acceptable limits for normal data[56]. All significance tests were two-tailed. Analyses of the ALSPAC, E-Risk, Dunedin and MCS cohorts were conducted using Stata v.17.0 (ref. 57), as well as Mplus v.8.2 for E-Risk[58]; analyses of WLS and HRS were conducted using R. Because E-Risk is a twin sample, we used structural equation models for dyads with indistinguishable members to take into account the unique structure of the data[59]. Because the MCS cohort has a complex stratified and clustered design and non-random dropout over the years, we used sampling weights that correct for design and nonresponse, as well as adjustment for clustering, following instructions published by the MCS Research Team[60].

In models predicting childhood and adolescent parenting, we adjusted for child sex. In models predicting parental investment to adult children, we also adjusted for parents' age, net worth (in WLS) or assets (in HRS), number of children, labour force status and, for analyses predicting help with childcare, we adjusted for physical proximity to offspring.

The exact n for each measure is reported in the measure description in the Supplementary Information; the sample sizes were chosen on the basis of the availability of valid data for each analysis (for example, valid data for polygenic scores and breastfeeding for analyses of the association between the two). We dealt with missing data in our analyses by only analysing participants who had valid data on all measures (constructed as described in the Supplementary Information). In ALSPAC, E-Risk, Dunedin and MCS we conducted sensitivity analyses using full information maximum likelihood estimation as implemented in Stata; this did not change the findings.

The premise and analysis plan for this project were preregistered at https://sites.duke.edu/moffittcaspiprojects/files/2021/07/Wertz_2019a.pdf (25 September 2019). Analyses reported here were checked for reproducibility by an independent data-analyst, who recreated the code by working from the manuscript and applied it to a fresh dataset.

## Reporting summary
Further information on research design is available in the Nature Portfolio Reporting Summary linked to this article.

## Data availability
MCS phenotypic data are available for free via the UK Data Service (https://ukdataservice.ac.uk/2020/10/14/millennium-cohort-study-age-17-data-now-available/); MCS genetic data are available for free, through managed access via the UCL CLS Data Access Committee (https://cls.ucl.ac.uk/data-access-training/data-access/accessing-data-directly-from-cls). ALSPAC phenotypic and genetic data are available for a fee, through managed access via the ALSPAC Executive Committee (http://www.bristol.ac.uk/alspac/researchers/access/). E-Risk and Dunedin phenotypic and genetic data are available for free, through managed access via the respective study units (https://sites.duke.edu/moffittcaspiprojects/data-use-guidelines/). HRS phenotypic and genetic polygenic-score data are available for free via the HRS study website (https://hrs.isr.umich.edu/data-products). WLS phenotypic data are available for free via the WLS study website (https://www.ssc.wisc.edu/wlsresearch/data/); WLS genetic data are available for free, through managed access via the WLS PIs (https://www.ssc.wisc.edu/wlsresearch/documentation/GWAS/).

## Code availability
The code for all analyses reported in the paper is available on request to the corresponding author (J.W.).

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

## Acknowledgements

This study uses data from the Dunedin and E-Risk cohorts, the ALSPAC cohort, the MCS cohort, the HRS cohort and the WLS cohort. The Dunedin and E-Risk cohorts are supported by grants from the National Institute on Aging (grant no. AG032282), the National Institute of Child Health and Development (grant nos. HD077482 and HD32948), the UK Medical Research Council (grant nos. MR/P005918/1 and G1002190) and the Jacobs Foundation. The Dunedin Research Unit is additionally supported by the New Zealand Health Research Council (programme grant no. 16-604) and New Zealand Ministry of Business, Innovation and Employment. The Dunedin Multidisciplinary Health and Development Research Unit at University of Otago is within the Ngāi Tahu tribal area who we acknowledge as first peoples, *tangata whenua* (translation, 'people of this land'). The ALSPAC cohort receives core support from the UK Medical Research Council and Wellcome Trust (grant no. 217065/Z/19/Z) and the University of Bristol. ALSPAC GWAS data were generated by Sample Logistics and Genotyping Facilities at Wellcome Sanger Institute and LabCorp (Laboratory Corporation of America) using support from 23andMe and funding by Wellcome Trust WT088806 (for mother GWAS data). A comprehensive list of ALSPAC funding is available on the ALSPAC website (http://www.bristol.ac.uk/alspac/external/documents/grant-acknowledgements.pdf). The MCS cohort is funded by grants from the Economic and Social Research Council. The HRS is supported by National Institute on Aging grant nos. U01AG009740, RC2AG036495 and RC4AG039029 and is conducted by the University of Michigan. The WLS is supported by National Institute on Aging grant nos. R01AG041868 and P30AG017266. The WLS genetic data are sponsored by the National Institute on Aging (grant nos. R01AG009775, R01AG033285 and R01AG041868) and was conducted by the University of Wisconsin. This work used a high-performance computing facility partially supported by grant no. 2016-IDG-1013 ('HARDAC+, Reproducible HPC for Next-Generation Genomics') from the North Carolina Biotechnology Center. We acknowledge the assistance of the Duke Molecular Physiology Institute Molecular Genomics Core for the generation of data for the paper. This research benefitted from GWAS results made publicly available by the SSGAC https://www.thessgac.org/. J.W. received support from an AXA Research Fund postdoctoral fellowship and from the Jacobs Foundation. A.D. is funded by the National Institute for Health Research (NIHR) Biomedical Research Centre at South London and Maudsley National Health Service Foundation Trust and King's College London. The views expressed are those of the authors and not necessarily those of the UK National Health Service or NIHR or any of the funders. We are extremely grateful to: the Dunedin Study members and their children, the Dunedin Unit research staff and Dunedin Study founder P. Silva; the E-Risk Study twins and their parents and the E-Risk Study team for their dedication, hard work and insights; the families who took part in the ALSPAC study, the midwives for their help in recruiting them and the whole ALSPAC team, which includes interviewers, computer and laboratory technicians, clerical workers, research scientists, volunteers, managers, receptionists and nurses; the MCS families for their time and cooperation, as well as the Centre for Longitudinal Studies (CLS), University College London (UCL) Social Research Institute, for the use of the MCS data and to the UK Data Service for making them available and the HRS and WLS study participants and study teams. Neither CLS nor the UK Data Service bear any responsibility for the analysis or interpretation of these data. This publication is the work of the authors and J.W. will serve as guarantor for the contents of this paper. The funders had no role in study design, data collection and analysis, decision to publish or preparation of the manuscript.

## Author contributions

J.W., T.E.M. and A.C. designed the research. J.W., T.E.M., A.C., A.D., M.B., J.C.B., H.L. and P.T.T. planned the study. T.E.M., A.C., R.P., L.A. and R.J.H. collected the data. J.W., J.C.B., D.L.C., H.L.H., H.L., K.S., P.T.T. and B.S.W. prepared the data for analysis. J.W., J.C.B., H.L., P.T.T., R.M.H. and S.L. analysed the data. J.W., T.E.M. and A.C. wrote the manuscript. All authors discussed the results, contributed to the revision of the manuscript and approved the final manuscript for submission.

## Competing interests

The authors declare no competing interests.

## Additional information

**Correspondence and requests for materials** should be addressed to Jasmin Wertz.

[1]Department of Psychology, University of Edinburgh, Edinburgh, UK. [2]Department of Psychology and Neuroscience, Duke University, Durham, NC, USA. [3]Social, Genetic and Developmental Psychiatry Centre, Institute of Psychiatry, Psychology and Neuroscience, King's College London, London, UK. [4]PROMENTA Research Center, University of Oslo, Oslo, Norway. [5]Center for Genomic and Computational Biology, Duke University, Durham, NC, USA. [6]Department of Psychiatry and Behavioral Sciences, Duke University, Durham, NC, USA. [7]School of Criminal Justice, University of Cincinnati, Cincinnati, OH, USA. [8]School of Psychology, Laval University, Quebec, Quebec, Canada. [9]Department of Genetics, Lineberger Comprehensive Cancer Center, University of North Carolina, Chapel Hill, NC, USA. [10]Department of Child and Adolescent Psychiatry, Institute of Psychiatry, Psychology and Neuroscience, King's College London, London, UK. [11]National and Specialist CAMHS Clinic for Trauma, Anxiety and Depression, South London and Maudsley NHS Foundation Trust, London, UK. [12]Department of Preventive and Social Medicine, University of Otago, Dunedin, New Zealand. [13]Institute for Interdisciplinary Data Science, University of Cincinnati, Cincinnati, OH, USA. [14]Dunedin Multidisciplinary Health and Development Research Unit, Department of Psychology, University of Otago, Dunedin, New Zealand. [15]Department of Psychology, University of Texas at Austin, Austin, TX, USA. [16]Population Research Center, University of Texas at Austin, Austin, TX, USA. ✉e-mail: jasmin.wertz@ed.ac.uk

# Reporting Summary

## Statistics

For all statistical analyses, confirm that the following items are present in the figure legend, table legend, main text, or Methods section.

| n/a | Confirmed | |
|---|---|---|
| ☐ | ☒ | The exact sample size (*n*) for each experimental group/condition, given as a discrete number and unit of measurement |
| ☐ | ☒ | A statement on whether measurements were taken from distinct samples or whether the same sample was measured repeatedly |
| ☐ | ☒ | The statistical test(s) used AND whether they are one- or two-sided *Only common tests should be described solely by name; describe more complex techniques in the Methods section.* |
| ☐ | ☒ | A description of all covariates tested |
| ☐ | ☒ | A description of any assumptions or corrections, such as tests of normality and adjustment for multiple comparisons |
| ☐ | ☒ | A full description of the statistical parameters including central tendency (e.g. means) or other basic estimates (e.g. regression coefficient) AND variation (e.g. standard deviation) or associated estimates of uncertainty (e.g. confidence intervals) |
| ☐ | ☒ | For null hypothesis testing, the test statistic (e.g. *F*, *t*, *r*) with confidence intervals, effect sizes, degrees of freedom and *P* value noted *Give P values as exact values whenever suitable.* |
| ☒ | ☐ | For Bayesian analysis, information on the choice of priors and Markov chain Monte Carlo settings |
| ☒ | ☐ | For hierarchical and complex designs, identification of the appropriate level for tests and full reporting of outcomes |
| ☒ | ☐ | Estimates of effect sizes (e.g. Cohen's *d*, Pearson's *r*), indicating how they were calculated |

*Our web collection on statistics for biologists contains articles on many of the points above.*

## Software and code

Policy information about availability of computer code

| Data collection | This study uses secondary data so we did not collect any data. |
|---|---|
| Data analysis | We did not use any software for collecting data. For constructing polygenic scores the softwares PRsice software v1.22, http://prsice.info/; Euesden, Lewis, & O'Reilly, 2015); LDPred (version 1.0.11) and PLINK v1.9 (Chang et al., 2015) were used. For analysing the data, Stata version 17.0 (StataCorp, 2021); Mplus version 8.2 (Muthén & Muthén, 1998-2017); R version 4.2.1 (2022-06-23). All the code is available on request. |

For manuscripts utilizing custom algorithms or software that are central to the research but not yet described in published literature, software must be made available to editors and reviewers. We strongly encourage code deposition in a community repository (e.g. GitHub). See the Nature Portfolio guidelines for submitting code & software for further information.

## Data

Policy information about availability of data

All manuscripts must include a data availability statement. This statement should provide the following information, where applicable:
- Accession codes, unique identifiers, or web links for publicly available datasets
- A description of any restrictions on data availability
- For clinical datasets or third party data, please ensure that the statement adheres to our policy

MCS phenotypic data are available for free via the UK Data Service (https://ukdataservice.ac.uk/2020/10/14/millennium-cohort-study-age-17-data-now-available/); MCS genetic data are available for free, through managed access via the UCL Centre for Longitudinal Studies Data Access Committee (https://cls.ucl.ac.uk/data-access-training/data-access/accessing-data-directly-from-cls/). ALSPAC phenotypic and genetic data are available for a fee, through managed access via the ALSPAC Executive Committee (http://www.bristol.ac.uk/alspac/researchers/access/). E-Risk and Dunedin phenotypic and genetic data are available for free, through managed access via the respective study units (https://sites.duke.edu/moffittcaspiprojects/data-use-guidelines/). HRS phenotypic and genetic polygenic-score data

# Field-specific reporting

Please select the one below that is the best fit for your research. If you are not sure, read the appropriate sections before making your selection.

☐ Life sciences   ☒ Behavioural & social sciences   ☐ Ecological, evolutionary & environmental sciences

For a reference copy of the document with all sections, see nature.com/documents/nr-reporting-summary-flat.pdf

# Behavioural & social sciences study design

All studies must disclose on these points even when the disclosure is negative.

| | |
|---|---|
| Study description | This study uses quantitative data. |
| Research sample | We used the following existing datasets: MCS cohort (n=6,732), based in the UK, nationally representative when weighted; ALSPAC cohort (n=7,588) based in the UK, not nationally representative; E-Risk cohort (n=880), based in the UK, nationally representative; Dunedin cohort (n=643), based in New Zealand, nationally representative; HRS cohort (n=8,652), based in the US, nationally representative; WRS cohort (n=8,479), based in the US, not nationally representative. Each sample was chose because it had measures of genetics in parents; measures of parenting; and a sample size of at least 200 individuals. Sources for the data are available in the links in the "Data availability" statement above. |
| Sampling strategy | We did not collect any data, so did not use a sampling strategy. In the original cohort recruitment, the following sampling strategies were used: MCS=stratified and clustered sampling; ALSPAC=opportunity sampling; E-Risk=stratified sampling; Dunedin=stratified sampling; HRS=stratified sampling; WLS=stratified sampling. Because we did not collect data, we did not predetermine sample sizes. However, we set out to include cohorts that would have at least 200 participants because of power calculations indicating that we would need at least n=200 individuals to be able to detect correlations of r=.20. All our sample sizes exceed this limit, often substantially so. |
| Data collection | We did not collect any data for this study. Instruments used in the original data collection were paper questionnaires, postal questionnaires, and online questionnaires. The interviewers collecting the data were blind to the study's hypotheses, in the sense that they were unaware of the study questions the data would be used for (because the data were collected before the research questions in this study had been developed). |
| Timing | For the MCS, data were collected between 2001 and 2019. For ALSPAC, data were collected between 1991 and 2010. For E-Risk, data were collected between 1998 and 2007. For Dunedin, data were collected between 1994 and 2019. For HRS, data were collected between 1992 and 2016. For WLS, data were collected between 1957 and 2011. |
| Data exclusions | In MCS and ALSPAC we excluded families with multiples (n=158 in MCS and n=184 in ALSPAC). Otherwise we did not make any exclusions of individuals who had data in our study variables. |
| Non-participation | For MCS, by age 17 (the last assessment age we included), n=8061 of the original sample had dropped out of the sample due to refusal to participate or inability to trace the participant. For ALSPAC, by age 17 (the last assessment age we included), n=7141 of the original sample had dropped out of the sample due to refusal to participate or inability to trace the participant. For E-Risk, by age 12 (the last assessment age we included), n=43 of the original n=1,116 had dropped out of the sample due to refusal to participate or inability to trace the participant. For Dunedin, there was only one assessment wave, so there was no dropout over time. For HRS and WLS, we used data from a minimum of one assesment wave, so people were included even if they dropped out at later assessment. |
| Randomization | This was not an experiment, so we did not assign people to experimental conditions (randomly or otherwise). |

# Reporting for specific materials, systems and methods

We require information from authors about some types of materials, experimental systems and methods used in many studies. Here, indicate whether each material, system or method listed is relevant to your study. If you are not sure if a list item applies to your research, read the appropriate section before selecting a response.

## Materials & experimental systems

| n/a | Involved in the study |
|-----|-----------------------|
| ☒ | ☐ Antibodies |
| ☒ | ☐ Eukaryotic cell lines |
| ☒ | ☐ Palaeontology and archaeology |
| ☒ | ☐ Animals and other organisms |
| ☐ | ☒ Human research participants |
| ☒ | ☐ Clinical data |
| ☒ | ☐ Dual use research of concern |

## Methods

| n/a | Involved in the study |
|-----|-----------------------|
| ☒ | ☐ ChIP-seq |
| ☒ | ☐ Flow cytometry |
| ☒ | ☐ MRI-based neuroimaging |

# Human research participants

Policy information about studies involving human research participants

| | |
|---|---|
| Population characteristics | This information is provided in the manuscript for each cohort. |
| Recruitment | This information is provided in the manuscript for each cohort. |
| Ethics oversight | This information is provided in the manuscript for each cohort. |

Note that full information on the approval of the study protocol must also be provided in the manuscript.

