## [Peer Review File · Nature Human Behaviour]

Peer Review Information

Journal: Nature Human Behaviour

Manuscript Title: Genetic associations with parental investment from conception to wealth inheritance in six cohorts

Corresponding author name(s): Jasmin Wertz

Reviewer Comments & Decisions:

Decision Letter, initial version:

29th March 2022

Dear Dr Wertz,

Thank you once again for your manuscript, entitled "Genetic influences on parental investment from conception to wealth inheritance: 30,000 parents in six cohorts," and for your patience during the peer review process.

Your manuscript has now been evaluated by 2 reviewers, whose comments are included at the end of this letter. Although the reviewers find your work to be of interest, they also raise some important concerns. We are interested in the possibility of publishing your study in Nature Human Behaviour, but would like to consider your response to these concerns in the form of a revised manuscript before we make a decision on publication.

To guide the scope of the revisions, the editors discuss the referee reports in detail within the team, including with the chief editor, with a view to (1) identifying key priorities that should be addressed in revision and (2) overruling referee requests that are deemed beyond the scope of the current study. In the case of your manuscript we ask that you address all reviewer comments in full, paying particular attention to ensuring that the framing and interpretation of your findings is robust and takes into account all concerns highlighted by our reviewers. We also agree with Reviewer 2 that the title of the manuscript should be amended to remove any implication of causal inference, and causal language should likewise be removed from the main text, since your results speak to associations, not causation. Please do not hesitate to get in touch if you would like to discuss these issues further.

Finally, your revised manuscript must comply fully with our editorial policies and formatting requirements. Failure to do so will result in your manuscript being returned to you, which will delay its consideration. To assist you in this process, I have attached a checklist that lists all of our requirements. If you have any questions about any of our policies or formatting, please don't hesitate

to contact me.

In sum, we invite you to revise your manuscript taking into account all reviewer and editor comments. We are committed to providing a fair and constructive peer-review process. Do not hesitate to contact us if there are specific requests from the reviewers that you believe are technically impossible or unlikely to yield a meaningful outcome.

We hope to receive your revised manuscript within two months. I would be grateful if you could contact us as soon as possible if you foresee difficulties with meeting this target resubmission date.

- Include a "Response to the editors and reviewers" document detailing, point-by-point, how you addressed each editor and referee comment. If no action was taken to address a point, you must provide a compelling argument. When formatting this document, please respond to each reviewer comment individually, including the full text of the reviewer comment verbatim followed by your response to the individual point. This response will be used by the editors to evaluate your revision and sent back to the reviewers along with the revised manuscript.
- Highlight all changes made to your manuscript or provide us with a version that tracks changes.

[REDACTED]

We look forward to seeing the revised manuscript and thank you for the opportunity to review your work. Please do not hesitate to contact me if you have any questions or would like to discuss these revisions further.

Sincerely,

Charlotte Payne

Charlotte Payne, PhD
Senior Editor
Nature Human Behaviour

Reviewer expertise:

Reviewer #1: behavioural genetics; academic achievement; intergenerational transmission

Reviewer #2: behavioural genetics; genetic nurture

REVIEWER COMMENTS:

Reviewer #1:

Remarks to the Author:

This manuscript was a pleasure to read. It is well written and interesting. Most research on parenting, or children's rearing environments, ignore genetics and jump to causal conclusions based on a phenotypic parent-child correlation. See e.g. Hart et al 2021.

In the current manuscript, across developmental stages and parenting measures, parents' EA-PGS predict parents' parenting behaviour.

In some analyses, this effect was reduced, but still significant, after controlling for parents' EA phenotype. In the interpretations that are given (Results and Discussion) it wasn't clear whether the fact that the effects were reduced was interpreted, or the fact that the effect was still significant. I think both parts of the finding are interesting and require interpretation.

In other analyses, the effect was reduced, but still significant, after controlling for children's EA-PGS. This effect shows parallels with the indirect genetic effect (or genetic nurture, or dynastic effects) on untransmitted alleles, albeit there the outcome measure is a trait in the offspring generation. Perhaps these parallels (and differences) could be spelt out. Does the effect here, on a trait in the parent generation, have a name in the literature? The effect is interpreted as an evocative gene-environment correlation. That seems plausible. But I was wondering whether it indeed gives direct evidence for evocative gene-environment correlations, or whether something else could (partly) explain this effect, like some of the limitations mentioned.

Regarding genetic nurture (Discussion, p.13, lines from 426), I'd like to point the authors to a very recent preprint by Nivard et al ([10.31234/osf.io/bhpm5](https://doi.org/10.31234/osf.io/bhpm5)). They show that the indirect genetic effect on childhood educational outcomes should not be interpreted as an effect of parenting, but as a family-wide effect (outside of the nuclear family). Would your findings also be consistent with a family-wide effect, rather than a parenting effect? Please discuss this in the manuscript.

Educational attainment shows substantial assortative mating (spousal correlation $\sim .40$). Does that affect the findings? (EA-)PGS include not only direct genetic effects, but also indirect genetic effects. Does that affect the findings? If so, please discuss so in the limitations section.

In the Introduction (p.4) the authors mention that most twin studies are of the type twins-as-children rather than twins-as-adults. A special type of twins-as-adults studies are the children-of-twin studies, which give both insights into the genetic and environmental influences on the parent characteristic (e.g., EA) and the genetic and environmental parent to child transmission. Introduction and/or Discussion, I feel that these studies need to be included. E.g. the review by McAdams et al 2014, or this recent MoBa study that included the effect of parental EA: Torvik et al 2020. What are the strengths and weaknesses of the authors' PGS approach compared to children-of-twin studies?

The study is on parenting and parental genetics, but the ultimate outcome would obviously be the

effects of parenting (and parental genetics) on offspring outcomes. The introduction and discussion indeed discuss the effects of parenting on offspring outcomes. However, offspring outcomes are not included in the study. By selling the manuscript, I feel that the authors are unconsciously trying to hide that, which makes the manuscript more difficult to follow. I would be explicit about this. The work is still very impressive!

References:

Hart, S. A., Little, C., & van Bergen, E. (2021). Nurture might be nature: cautionary tales and proposed solutions. *NPJ science of learning*, 6(1), 1-12.

McAdams, T. A., Neiderhiser, J. M., Rijdsdijk, F. V., Narusyte, J., Lichtenstein, P., & Eley, T. C. (2014). Accounting for genetic and environmental confounds in associations between parent and child characteristics: a systematic review of children-of-twins studies. *Psychological bulletin*, 140(4), 1138.

Torvik, F. A., Eilertsen, E. M., McAdams, T. A., Gustavson, K., Zachrisson, H. D., Brandlistuen, R., ... & Ystrom, E. (2020). Mechanisms linking parental educational attainment with child ADHD, depression, and academic problems: a study of extended families in The Norwegian Mother, Father and Child Cohort Study. *Journal of Child Psychology and Psychiatry*, 61(9), 1009-1018.

Nivard, M. G., Belsky, D., Harden, K. P., Baier, T., Ystrom, E., & Lyngstad, T. H. (2022, February 22). Neither nature nor nurture: Using extended pedigree data to elucidate the origins of indirect genetic effects on offspring educational outcomes. <https://doi.org/10.31234/osf.io/bhpm5>

Reviewer #2:

Remarks to the Author:

MANUSCRIPT REVIEW

Journal: Nature: Human Behaviour

Manuscript Number: NATHUMBEHAV- 22020384

Title: Genetic influences on parental investment from conception to wealth inheritance: 30,000 parents in six cohorts

This is an interesting and well written paper that addresses a question of considerable interest – To what extent are genetic effects associated with educational attainment correlated with being raised in advantageous circumstances? As explained in the paper, the existence of gene-environment correlation would have significant implications for research on genetic and environmental contributions to academic achievement as it would indicate that the two effects are intertwined, making it difficult to assess their separate contributions.

There are many strengths to the paper. Results are clearly presented and cautiously interpreted. The study was pre-registered and takes advantage of several large and informative samples. Multiple indicators of parenting and multiple developmental periods are investigated, allowing the researchers to engage in what I would consider to be more speculative yet interesting analyses about lifetime

cumulative effects. And the investigators use a relatively novel approach to investigating gene-environment correlation based on polygenic scores (PGS) for educational attainment (EA). I really don't have much to say by way of criticism of the paper. My only question is whether the results reported, which I think are largely anticipated by the extensive literature linking parental (phenotypic) educational attainment with parenting, rise to the level justifying publication in your journal.

General Comment:

Although other results are reported (e.g., father versus mother effects), in my judgment the major findings from the study come from two interrelated sets of analyses. In the first, the investigators determined the extent to which maternal PGS for educational attainment was associated with various indicators of parenting (e.g., Figure 2). Results were generally consistent across the separate studies in showing that maternal PGS was positively associated with advantageous parenting behaviors (e.g., cognitive stimulation, academic assistance) and negatively associated with disadvantageous parental behaviors (e.g., maintaining a chaotic home). Notably, this pattern of association was observed at multiple developmental stages (e.g., maternal PGS was negatively associated with prenatal drinking and smoking, Figure 1). Although findings from this first stage of the analysis constituted critical justification for the second stage of analysis, in my opinion they are not particularly surprising. That is, we know that maternal educational attainment is positively associated with providing a cognitive stimulating home, maternal warmth, academic support, breast feeding, etc. Consequently, it would be surprising if maternal PGS, which after all is derived to predict maternal educational attainment, did not show a similar pattern of association.

The second set of analyses arguably provides the most novel findings from the study. In the second stage (only possible for one developmental period – childhood – because that was the only stage for which child genotypes were available), the investigators determined whether adjusting for the child's EA PGS statistically accounted for the associations of maternal PGS with parenting indicators (Figure 3). To understand the significance of adjusting for child PGS it is perhaps helpful to distinguish between what behavioral geneticists term passive and reactive gene-environment correlation (r_{ge}). Passive r_{ge} arises because parents who transmit genes to their children also help construct their rearing environments, so that the two (genes and environments) become correlated through no direct involvement of the child (i.e., passively). Alternatively, parenting behaviors like providing cognitive stimulation may be a reaction to the child's cognitive talents and if cognitive talents are in part heritable, this will lead to a correlation between the environment the child experiences and the genes they inherited from their parents.

In principle, correcting for child PGS should eliminate this second pathway to r_{ge}, so that the absence of a maternal PGS effect after correction for child PGS might lead to the conclusion that it is the reactive r_{ge} pathway that is driving the observed gene-environment correlation. Although this is a reasonable conclusion, in my opinion it comes with two major caveats. First, a non-null maternal PGS effect after correction is ambiguous. It cannot be used to conclude that reactive processes are not at play as a non-null effect is compatible with gene-environment correlation being due to some combination of passive and reactive processes. Consequently, in my opinion it is only null effects are potentially informative. Yet concluding that the maternal PGS effect after correction is null because it is no longer statistically significant comes with all the difficulties associated with accepting the null hypothesis under this statistical paradigm (e.g., power).

In any case, in the results summarized in Figure 3, maternal PGS is non-significantly associated after correction with only one of the parenting behaviors, Parental Monitoring. By my logic, this would lead to the conclusion that we cannot be certain why the other 5 parenting behaviors are correlated with maternal PGS but that we might conclude that the association of parental monitoring was driven by child effects. The latter is consistent with a larger literature on parental monitoring, so, caveats aside, the conclusion seems reasonable to me.

Specific Comments:

1. The title of the paper claims that the study investigated "genetic influences". Nonetheless, the only methodology used involves correlating PGS for educational attainment with various environmental indicators. It is not clear to me how causality is being established here and it seems to me the abstract is more accurate when it characterizes the findings as revealing "widespread associations between parental genetics . . . and parental behavior".
2. As best I could tell, there is no where in the paper where the investigators discuss the limitations of drawing conclusions about the genetics of educational attainment from a PGS that accounts for only a small portion of the observed (and likely heritable) variance in educational attainment.
3. On page 9 of the paper, the investigators state that "Children's polygenic scores were associated with most parenting measures, suggesting the presence of child effects." I don't see how the children's PGS are informative as they are necessarily correlated with parent PGS and so should be correlated with the same indicators the parent PGS are correlated with under any model of gene-environment correlation.
4. I confess that I am a bit confused by the results reported in Supplementary Table S1. That table reports the association of parental EA PGS on the parenting indicators after parent EA has been statistically controlled. Given that the PGS were derived to predict EA, wouldn't we expect the adjusted values to be null, apart from sampling fluctuation and perhaps unique features of the original GWAS. I am not sure what to make of non-null findings here.

I am in the habit of signing my reviews and so authorize the journal editor to transmit my signed comments to the authors.

Respectfully submitted,

Matt McGue, Ph.D.

Author Rebuttal to Initial comments

Reviewer #1:

We thank reviewer 1 for their comments and suggested improvements to our manuscript.

This manuscript was a pleasure to read. It is well written and interesting. Most research on parenting, or children's rearing environments, ignore genetics and jump to causal conclusions based on a phenotypic parent-child correlation. See e.g. Hart et al 2021.

We thank the reviewer for these nice comments and for referencing this article, which we now cite in the Introduction (page 4).

- 1) *In the current manuscript, across developmental stages and parenting measures, parents' EA-PGS predict parents' parenting behaviour. In some analyses, this effect was reduced, but still significant, after controlling for parents' EA phenotype. In the interpretations that are given (Results and Discussion) it wasn't clear whether the fact that the effects were reduced was interpreted, or the fact that the effect was still significant. I think both parts of the finding are interesting and require interpretation.*

We are grateful for the opportunity to clarify. In the Results, we now specifically mention that we are interpreting the *reduction* in the effect: "The reduction in the size of genetic associations suggests at least two possibilities." (page 10). In the Discussion, we now discuss both the reduction in the effect, as well as the finding that some associations remain significant even after controlling for parental educational attainment. This section now reads: "Associations between parental genetics and parental investment reduced substantially when parental educational attainment was taken into account, though some associations remained significant. The finding that associations were reduced indicates that links between parental polygenic scores and parenting are largely due to factors associated with parental educational attainment. For example, parental investment requires resources (e.g. money to buy books or healthy foods, or to leave as an inheritance) and specific knowledge (e.g. knowing how to support schooling) that parents with higher polygenic scores may have been able to acquire through their education (Davis-Kean, Tighe, & Waters, 2021). Alternatively, parents with higher education polygenic scores may have personal characteristics (e.g. cognitive and self-control skills) that predict both educational attainment as well as greater parental investment (D. W. Belsky et al., 2016). The finding that some associations remained significant indicates that even net of their completed education, parents who differ in their polygenic scores differ in their parenting (though residual effects were small). This suggests that the education polygenic score captures personal characteristics that predict individuals' behaviour over and above educational attainment (Belsky et al., 2016). However, it could also reflect sampling or measurement differences between the original GWAS and our cohorts." (page 14).

- 2) In other analyses, the effect was reduced, but still significant, after controlling for children's EA-PGS. This effect shows parallels with the indirect genetic effect (or genetic nurture, or dynastic effects) on untransmitted alleles, albeit there the outcome measure is a trait in the offspring generation. Perhaps these parallels (and differences) could be spelt out. Does the effect here, on

a trait in the parent generation, have a name in the literature? The effect is interpreted as an evocative gene-environment correlation. That seems plausible. But I was wondering whether it indeed gives direct evidence for evocative gene-environment correlations, or whether something else could (partly) explain this effect, like some of the limitations mentioned.

We thank the reviewer for this comment. Indirect genetic effects are effects of the genotype of one individual on the phenotype of other individuals. For example, prior research shows that parental genotype is associated with child educational attainment over and above child genotype (i.e. over and above genetic transmission from parent to child) (Kong et al., 2018). As the reviewer points out, a difference between this and our research is that we do not look at child outcomes. A parallel is that our research suggests specific aspects of parental behaviour across development that could mediate indirect genetic effects. We have revised our Discussion section to spell this out more explicitly: “Second, molecular-genetic studies of parent-offspring trios show that genes that are not passed on from parents to offspring still affect offspring outcomes (Bates et al., 2018; Kong et al., 2018). This finding suggests that genes influence parental characteristics in ways that affect offspring development, a phenomenon referred to as “genetic nurture” (Young, Benonisdottir, Przeworski, & Kong, 2019). Although our study does not look at child outcomes, our analyses point to specific aspects of parental behaviour across development that could mediate these effects.” (page 13).

The effect of child genotype on parenting net of parent genotype is usually referred to as an evocative gene-environment correlation; this is the name we use as well. Evocative gene-environment correlation arises if an individual’s genetic make-up evokes certain responses from their environment. A limitation that we outline in the Discussion is that the association between children’s polygenic score and parenting net of mothers’ polygenic scores could partly reflect father’s genetics (because fathers’ polygenic score is not controlled for in the model) (page 13).

The effect of parent genotype on parenting does not have an established name, but genetic influences on parenting are sometimes referred to as “the nature of nurture” (e.g. Plomin & Bergeman, 1991), so we are now using this phrase in the manuscript (page 13).

- 3) Regarding genetic nurture (Discussion, p.13, lines from 426), I’d like to point the authors to a very recent preprint by Nivard et al (10.31234/osf.io/bhpm5). They show that the indirect genetic effect on childhood educational outcomes should not be interpreted as an effect of parenting, but as a family-wide effect (outside of the nuclear family). Would your findings also be consistent with a family-wide effect, rather than a parenting effect? Please discuss this in the manuscript.

We thank the reviewer for pointing out this paper. The Nivard et al. paper shows that the indirect genetic effect of parent genotype on offspring educational attainment becomes statistically nonsignificant once the genotype of parents’ siblings is controlled for. What this result suggests is that

the processes that mediate indirect genetic effects are not specific to one nuclear family, but shared across extended families, including siblings and grandparents. The paper refers to this as a “family-wide” effect.

Our findings are consistent with these findings, because we observe genetic influences on parenting. Genetic influences on a phenotype (here parenting) imply that the phenotype will to some extent be shared across genetically-related family members. For example, because siblings share genes with each other, and genes are associated with parenting, siblings are likely to resemble each other in how they parent their offspring, more so than non-siblings. Likewise, because parents share genes with their own parents, they are likely to resemble each other in their parenting. This means that parenting itself “runs in families” and may therefore contribute to what the Nivard et al. paper refers to as “dynastic [i.e. family-wide] stratification in environments relevant to success in school”.

It may seem surprising that the findings of our paper are consistent with the Nivard et al., paper, because that paper refers to parenting as a “within-family process” and contrasts it with “dynastic [i.e. family-wide] stratification”. This description suggests that parenting behaviour is somewhat idiosyncratic to a nuclear family (a within-family process), rather than shared across extended family members (a family-wide process). However, as explained above, the presence of genetic influences on parenting indicates that parenting is not exclusively a within-family process but instead is at least to some extent shared across related family members (including siblings and grandparents).

We now briefly mention this in the paper (page 13): “[...] Together, these findings provide evidence that parenting is partly heritable, which is sometimes referred to as the “nature of nurture” (Plomin & Bergeman, 1991). Alongside environmental influences, genetic influences may partly explain why parenting tends to “runs in families”, i.e. why there tends to be resemblance in parenting behaviours across siblings and across generations (Belsky, Conger, & Capaldi, 2009)”. We kept this Discussion brief because the Nivard et al. paper is currently a preprint, and the descriptions of these processes may change in the finalised version, so we did not want to characterise the paper in a way that will differ from its final published version (this is also why we did not cite the preprint here, though we are happy to do so if the reviewer and editor think this would be appropriate).

- 4) Educational attainment shows substantial assortative mating (spousal correlation $\sim .40$). Does that affect the findings? (EA-)PGS include not only direct genetic effects, but also indirect genetic effects. Does that affect the findings? If so, please discuss so in the limitations section.

If there is assortative mating, it would mean that what looks like an association between mothers’ genetics and parenting could partly reflect associations between fathers’ genetics and parenting. We note that this would still be consistent with the overall conclusion of our paper, which is that parental genetics are associated with parenting. We now mention this limitation on page 13: “[...] Furthermore, to the extent that mothers’ and fathers’ genes are correlated, e.g. because of assortative mating (Young

et al., 2019), associations between mothers' genetics and parenting could partly reflect fathers' genetics."

- 5) In the Introduction (p.4) the authors mention that most twin studies are of the type twins-as-children rather than twins-as-adults. A special type of twins-as-adults studies are the children-of-twin studies, which give both insights into the genetic and environmental influences on the parent characteristic (e.g., EA) and the genetic and environmental parent to child transmission. Introduction and/or Discussion, I feel that these studies need to be included. E.g. the review by McAdams et al 2014, or this recent MoBa study that included the effect of parental EA: Torvik et al 2020. What are the strengths and weaknesses of the authors' PGS approach compared to children-of-twin studies?

We fully agree with the reviewer about the value of children-of-twin studies. We now refer to them more explicitly in the Introduction (page 4). A strength of our approach is that we were able to use larger and more population-representative samples with more extensive data on parenting across the lifespan, than has been possible in children-of-twin studies (e.g. the McAdams et al. 2014 meta-analysis cites six existing children-of-twin cohorts, few with sample sizes of $n > 1000$; even fewer with parenting data across offspring ages). A weakness of our approach, which we now mention explicitly in the Limitations (page 13) is that polygenic scores capture only a limited portion of the entire genetic influence on a phenotype, compared to the children-of-twin design.

- 6) The study is on parenting and parental genetics, but the ultimate outcome would obviously be the effects of parenting (and parental genetics) on offspring outcomes. The introduction and discussion indeed discuss the effects of parenting on offspring outcomes. However, offspring outcomes are not included in the study. By selling the manuscript, I feel that the authors are unconsciously trying to hide that, which makes the manuscript more difficult to follow. I would be explicit about this. The work is still very impressive!

We thank the reviewer for this suggestion, and we are now more explicit about the fact that we are not analysing offspring outcomes, e.g. in the Discussion (page 13; "Although our study does not look at child outcomes, our analyses point to specific aspects of parental behaviour across development that could mediate these effects."). We note that we have tested parenting measures that have been shown to be powerful socialization predictors of offspring cognitive, educational, behavioral, and health outcomes.

Reviewer #2:

We thank reviewer 2 for their comments and suggested improvements to our manuscript.

This is an interesting and well written paper that addresses a question of considerable interest – To

what extent are genetic effects associated with educational attainment correlated with being raised in advantageous circumstances? As explained in the paper, the existence of gene-environment correlation would have significant implications for research on genetic and environmental contributions to academic achievement as it would indicate that the two effects are intertwined, making it difficult to assess their separate contributions.

There are many strengths to the paper. Results are clearly presented and cautiously interpreted. The study was pre-registered and takes advantage of several large and informative samples. Multiple indicators of parenting and multiple developmental periods are investigated, allowing the researchers to engage in what I would consider to be more speculative yet interesting analyses about lifetime cumulative effects. And the investigators use a relatively novel approach to investigating gene-environment correlation based on polygenic scores (PGS) for educational attainment (EA). I really don't have much to say by way of criticism of the paper.

We thank reviewer 2 for these nice comments.

- 1) My only question is whether the results reported, which I think are largely anticipated by the extensive literature linking parental (phenotypic) educational attainment with parenting, rise to the level justifying publication in your journal.

The reviewer states that the reported results are largely anticipated, and may therefore not rise to the level justifying publication in this journal. We have two comments. First, a premium placed on surprising results is thought to have contributed to psychology's reproducibility crisis (Nosek et al., 2022). Second, while the reviewer may have anticipated the results, most readers and researchers are unlikely to be as knowledgeable about this research area as Professor McGue. For example, as Reviewer 1 notes: "*Most research on parenting, or children's rearing environments, ignore genetics and jump to causal conclusions based on a phenotypic parent-child correlation. See e.g. Hart et al 2021.*". As another example, the first author has now presented these findings to several interdisciplinary audiences, most recently at a meeting attended by experts in demography, sociology and economics. Listeners found the findings novel and thought-provoking, leading us to think that the interdisciplinary readership of the journal may have a similar reaction.

- 2) Although other results are reported (e.g., father versus mother effects), in my judgment the major findings from the study come from two interrelated sets of analyses. In the first, the investigators determined the extent to which maternal PGS for educational attainment was associated with various indicators of parenting (e.g., Figure 2). Results were generally consistent across the separate studies in showing that maternal PGS was positively associated with advantageous parenting behaviors (e.g., cognitive stimulation, academic assistance) and negatively associated with disadvantageous parental behaviors (e.g., maintaining a chaotic

home). Notably, this pattern of association was observed at multiple developmental stages (e.g., maternal PGS was negatively associated with prenatal drinking and smoking, Figure 1). Although findings from this first stage of the analysis constituted critical justification for the second stage of analysis, in my opinion they are not particularly surprising. That is, we know that maternal educational attainment is positively associated with providing a cognitive stimulating home, maternal warmth, academic support, breast feeding, etc. Consequently, it would be surprising if maternal PGS, which after all is derived to predict maternal educational attainment, did not show a similar pattern of association.

We would like to briefly comment on the reviewer's point that the "findings from this first stage of the analysis [...] are not particularly surprising". First, our replies to the reviewer's earlier comment (comment #1) apply to this comment as well (i.e. the potential issues with placing a premium on surprising findings; and the point that these findings may not be as anticipated for everyone as they are for the reviewer). Second, although we fully agree with the reviewer that the association between education-associated genetics and parenting could be anticipated based on prior research findings, we still think there is value in testing these associations. For one, the finding of a phenotypic association between education and parenting does not in itself mean that there must also be a genetic association between the two, especially given the fact that the polygenic score only explains a fraction of the heritability of educational attainment. Furthermore, the implications of a gene-environment correlation go beyond those of a phenotypic correlation between education and parenting. A gene-environment correlation would imply that a) genes can have environmentally-mediated effects via parenting; b) the unique contributions of genes and parenting cannot be easily separated (as the reviewer pointed out); and b) the advantage of inheriting education-associated genes is associated with the advantage of invested parenting, resulting in a 'double whammy' of favourable genes and favourable environments. Although previous studies have reported links between education-associated genes and some aspects of parenting during children's early years, here we show that these links are not unique to a few aspects of parenting in the early years of life, but are widespread across measures of parental investment, and that they continue and accumulate into offspring adulthood.

- 3) The second set of analyses arguably provides the most novel findings from the study. In the second stage (only possible for one developmental period – childhood – because that was the only stage for which child genotypes were available), the investigators determined whether adjusting for the child's EA PGS statistically accounted for the associations of maternal PGS with parenting indicators (Figure 3). To understand the significance of adjusting for child PGS it is perhaps helpful to distinguish between what behavioral geneticists term passive and reactive gene-environment correlation (r_{ge}). Passive r_{ge} arises because parents who transmit genes to their children also help construct their rearing environments, so that the two (genes and environments) become correlated through no direct involvement of the child (i.e., passively). Alternatively, parenting behaviors like providing cognitive stimulation may be a reaction to the child's cognitive talents and if cognitive talents are in part heritable, this will lead to a correlation between the environment the child experiences and the genes they inherited from

their parents. In principle, correcting for child PGS should eliminate this second pathway to rge, so that the absence of a maternal PGS effect after correction for child PGS might lead to the conclusion that it is the reactive rge pathway that is driving the observed gene-environment correlation. Although this is a reasonable conclusion, in my opinion it comes with two major caveats. First, a non-null maternal PGS effect after correction is ambiguous. It cannot be used to conclude that reactive processes are not at play as a non-null effect is compatible with gene-environment correlation being due to some combination of passive and reactive processes. Consequently, in my opinion it is only null effects are potentially informative. Yet concluding that the maternal PGS effect after correction is null because it is no longer statistically significant comes with all the difficulties associated with accepting the null hypothesis under this statistical paradigm (e.g., power). In any case, in the results summarized in Figure 3, maternal PGS is non-significantly associated after correction with only one of the parenting behaviors, Parental Monitoring. By my logic, this would lead to the conclusion that we cannot be certain why the other 5 parenting behaviors are correlated with maternal PGS but that we might conclude that the association of parental monitoring was driven by child effects. The latter is consistent with a larger literature on parental monitoring, so, caveats aside, the conclusion seems reasonable to me.

We agree with the reviewer's conclusion that our findings suggest that the association between the polygenic score and parental monitoring is driven by child effects. As the reviewer points out, this is consistent with a larger literature on parental monitoring and is, in fact, also consistent with our own prior findings in the E-Risk cohort (using a twin-design). We have added a sentence to the Results section acknowledging these findings (page 9): "This finding is consistent with prior research reporting child effects on parental monitoring (Stattin & Kerr, 2000; Wertz et al., 2016)."

- 4) The title of the paper claims that the study investigated "genetic influences". Nonetheless, the only methodology used involves correlating PGS for educational attainment with various environmental indicators. It is not clear to me how causality is being established here and it seems to me the abstract is more accurate when it characterizes the findings as revealing "widespread associations between parental genetics . . . and parental behavior".

We have changed the title to remove any implication of a causal effect. It now reads: "Genetic associations with parental investment from conception to wealth inheritance in six cohorts". We have also removed causal language from the manuscript (all instances of "genetic influences on.." or "genes affecting.." were changed to "genetic associations with").

- 5) As best I could tell, there is nowhere in the paper where the investigators discuss the limitations of drawing conclusions about the genetics of educational attainment from a PGS that accounts for only a small portion of the observed (and likely heritable) variance in educational attainment.

We have added this limitation to the Discussion (page 13). It reads: “Seventh, although the polygenic score used in our study accounts for a considerable portion of variance in educational attainment (approx 10%; Lee et al., 2018); this is slightly lower than the estimated SNP heritability (approx. 15%; Lee et al., 2018) and far lower than the estimated twin heritability (approx 40%; Branigan, McCallum, & Freese, 2013). Our effect sizes may change with future iterations of the polygenic score that capture more heritability.” (

- 6) On page 9 of the paper, the investigators state that “Children’s polygenic scores were associated with most parenting measures, suggesting the presence of child effects.” I don’t see how the children’s PGS are informative as they are necessarily correlated with parent PGS and so should be correlated with the same indicators the parent PGS are correlated with under any model of gene-environment correlation.

We agree this could be confusing and have deleted part of this sentence (“suggesting the presence of child effects”) so that the focus is on the more informative model that includes both maternal and child polygenic score (page 9).

- 7) I confess that I am a bit confused by the results reported in Supplementary Table S1. That table reports the association of parental EA PGS on the parenting indicators after parent EA has been statistically controlled. Given that the PGS were derived to predict EA, wouldn’t we expect the adjusted values to be null, apart from sampling fluctuation and perhaps unique features of the original GWAS. I am not sure what to make of non-null findings here.

We agree that one would expect the association between polygenic score and parenting to be reduced when controlling for phenotypic educational attainment. We interpret the non-null result to reflect either sampling or measurement differences between the original GWAS and our cohorts, as mentioned by the reviewer, or genuine effects of parent genetics on parenting that are not mediated by parental educational attainment. Not everyone with a high polygenic score gets a lot of education, and some people get more education than their polygenic score would have predicted. Furthermore, previous research suggests that the education polygenic score captures personal characteristics that predict individuals’ behaviour over and above educational attainment (e.g. Belsky et al., 2016). We now explain this better in the paper (page 14): “The finding that some associations remained significant indicates that even net of their completed education, parents who differ in their polygenic scores differ in their parenting (though residual effects were small). This suggests that the education polygenic score captures personal characteristics that predict individuals’ behaviour over and above educational attainment (Belsky et al., 2016). However, it could also reflect sampling or measurement differences between the original GWAS and our cohorts.” (page 14).

References

Belsky, D. W., Moffitt, T. E., Corcoran, D. L., Domingue, B., Harrington, H., Hogan, S., ... Caspi, A. (2016).

The genetics of success: How single-nucleotide polymorphisms associated with educational attainment relate to life-course development. *Psychological Science*, 27, 957–972.

<https://doi.org/10.1177/0956797616643070>

Kong, A., Thorleifsson, G., Frigge, M. L., Vilhjalmsón, B. J., Young, A. I., Thorgeirsson, T. E., ... Stefansson, K. (2018). The nature of nurture: Effects of parental genotypes. *Science*, 359, 424–428.

<https://doi.org/10.1126/science.aan6877>

Nivard, M., Belsky, D., Harden, K. P., Baier, T., Ystrom, E., & Lyngstad, T. H. (2022). Neither nature nor nurture: Using extended pedigree data to elucidate the origins of indirect genetic effects on offspring educational outcomes. [Preprint.] February 22, 2022 [accessed 2022 June 13]. Available from: [10.31234/osf.io/bhpm5](https://doi.org/10.31234/osf.io/bhpm5)

Nosek, B. A., Hardwicke, T. E., Moshontz, H., Allard, A., Corker, K. S., Dreber, A., ... Vazire, S. (2022).

Replicability, Robustness, and Reproducibility in Psychological Science. *Annual Review of Psychology*, 73, 719–748. <https://doi.org/10.1146/ANNUREV-PSYCH-020821-114157>

Decision Letter, first revision:

26th September 2022

Dear Dr Wertz,

Thank you once again for your manuscript, entitled "Genetic associations with parental investment from conception to wealth inheritance in six cohorts," and for your patience during the peer review process.

Your manuscript has now been evaluated by 3 reviewers, whose comments are included at the end of this letter. As you will see, one of the reviewers (Reviewer 1) also reviewed the paper in the previous round. Unfortunately Reviewer 2 from the previous round could not re-review, so we recruited two further reviewers (Reviewers 3 and 4) who have overlapping expertise with Reviewer 2. Although the reviewers find your work to be of interest, they also raise some important concerns. We are interested in the possibility of publishing your study in *Nature Human Behaviour*, but would like to consider your response to these concerns in the form of a revised manuscript before we make a decision on publication.

To guide the scope of the revisions, the editors discuss the referee reports in detail within the team, including with the chief editor, with a view to (1) identifying key priorities that should be addressed in revision and (2) overruling referee requests that are deemed beyond the scope of the current study.

We hope that you will find the prioritised set of referee points to be useful when revising your study. Please do not hesitate to get in touch if you would like to discuss these issues further.

1) Please add a Figure clearly outlining the causal pathways examined in your analyses, as requested by Reviewer 1.

2) Please run your analyses with the adjustments suggested by Reviewer 2, and report these in your SI or main manuscript.

2) We ask that in this round you ensure that you incorporate reviewer suggestions in full in the manuscript text itself, including their concerns about key limitations of the work and how it relates to other literature in the field, including the preprint mentioned by Reviewer 1 (please note that when referring to preprints, these should be described as 'a non-peer-reviewed preprint' to indicate that they have not been peer reviewed).

In sum, we invite you to revise your manuscript taking into account all reviewer and editor comments. We are committed to providing a fair and constructive peer-review process. Do not hesitate to contact us if there are specific requests from the reviewers that you believe are technically impossible or unlikely to yield a meaningful outcome.

We hope to receive your revised manuscript within two months. I would be grateful if you could contact us as soon as possible if you foresee difficulties with meeting this target resubmission date.

- Include a "Response to the editors and reviewers" document detailing, point-by-point, how you addressed each editor and referee comment. If no action was taken to address a point, you must provide a compelling argument. When formatting this document, please respond to each reviewer comment individually, including the full text of the reviewer comment verbatim followed by your response to the individual point. This response will be used by the editors to evaluate your revision and sent back to the reviewers along with the revised manuscript.
- Highlight all changes made to your manuscript or provide us with a version that tracks changes.

[REDACTED]

We look forward to seeing the revised manuscript and thank you for the opportunity to review your work. Please do not hesitate to contact me if you have any questions or would like to discuss these revisions further.

Sincerely,

Charlotte Payne

Charlotte Payne, PhD
Senior Editor
Nature Human Behaviour

REVIEWER COMMENTS:

Reviewer #1:
Remarks to the Author:

Thank you for the response letter and the revised manuscript with tracked changes (minor thing: page numbers in the manuscript would have been helpful). I still like the work and the manuscript, but in general, I feel that the authors have mostly responded to the reviewers' comments in the letter but haven't done so much with incorporating our concerns in revising the manuscript. In particular, I'd like to see the following concerns better incorporated into the manuscript:

- I think the preprint by Nivard et al. is very relevant for the current manuscript and discussion on whether parental EA-genetics indicate the importance of parenting. It feels like the authors think so too, given the full-page discussion of this work in the response letter. It seems logical to me to reflect on this work in relation to their work in the manuscript. Yes, because of the heritability of parenting phenotypes, siblings will resemble each other in how they parent their offspring. This indeed fits with Nivard et al's finding that the average sib-ship's PGS/PGI in the parent generation explains variance in the offspring's educational outcomes. But in the following analysis, Nivard et al find "find no evidence that parents' PGI are specifically related to offspring academic achievement over and above the average PGI of the siblings in the parental generation."
- As I said previously, the ultimate outcome would be the effect of parenting (and parental genetics) on offspring outcomes. In response to my comment, the authors have only added less than half of a sentence towards the end of the manuscript ("although our study does not look at child outcomes"). That doesn't solve the problem of not being clear and open about what in the causal chain is and is not studied in the current manuscript. I'd suggest that the authors open up, perhaps illustrated in a figure in the introduction (a DAG?), about which part(s) of the causal chain between parenting and child outcomes are tested here and which are assumed based on the literature. Regarding the discussion of that literature, it helps if it's clear to the reader what in the literature (on associations or effects of parenting on child outcomes) is merely correlational and what has been demonstrated as causal (or somewhere in between). Not all paths can be tested in one study, but it helps the reader to

follow the current work and how that fits in the bigger picture.

- I feel Reviewer's 2 concerns too are discussed at length in the response letter, but not so much incorporated in the manuscript.

Reviewer #3:
Remarks to the Author:

This paper is a good addition to the literature about genetic influences on environmental measures ("nature of nurture"). Although "genetic nurture" is a well-established phenomenon (Kong et al., 2018, Willoughby et al., 2021), this study is valuable for its large sample size and the wide array of parenting traits it explores. I also liked the idea of controlling for offspring PGS as a way to approximate evocative rGE. A few comments:

- 1) I think the main weakness of the study is that it only looks at maternal PGS, while examining paternal influences only as a sort of sensitivity analysis. For studies where both mothers and fathers are available, wouldn't it make more sense to use mid-parent PGS instead of only maternal PGS? This would also address concerns about the role of assortative mating. Parenting is usually a two-person job, and examining only maternal influence may tell only half the story.
- 2) As described by Table 1, childhood periods are covered by NZ and UK cohorts, while adulthood is covered by US cohorts. This is a potential source of bias, is it clearly acknowledged in the paper?
- 3) The authors report that "mothers with higher education polygenic scores were also less likely to drink heavily during pregnancy". It looks like this association is only significant in ALSPAC but not in MCS, so this should be acknowledged.
- 4) In the analysis of intergenerational supports, the model controls for "respondent's age, sex, assets/net worth, number of children, labour force status, year/wave, and proximity". While some of the controls make sense (age, sex, number of children, year/wave, maybe proximity), the rest, especially assets and labour force status are clearly potential mediators of the association. Controlling for mediators may underestimate the effect that we are interested in (see the classic Meehl 1971: "High school yearbooks: a reply to Schwarz"). This may be responsible for the null association with wealth inheritance in the WLS.
- 5) The composite of parenting behaviours across time was only used in the E-Risk cohort, apparently because of its low attrition rate. Wouldn't the power of the study be boosted by including all cohorts, perhaps using weights to account for differential attrition between them?
- 6) Two minor observations: 1) in line 357, there is an in-text reference. 2) in line 431 the authors refer to "the most recent GWAS of educational attainment" – this had been corrected in the introduction, it needs to be corrected there as well.

Reviewer #4:

Remarks to the Author:

This is a very well-written paper that was a pleasure to read. It is also very well-structured and straightforward. I also think it is an important paper for the field which links genetics and research on parenting. Reviewers from the previous round raised crucial questions, and I see the authors addressed these sufficiently and convincingly. I do not have much to add to those – the manuscript has been improved making it hard to criticise.

My only question is about the polygenic score for educational attainment. While authors use it as the main analytic tool (which is meaningful), we know that this polygenic score has high genetic correlations with other outcomes. For example, as Mills et al. 2021 show the genetic correlation between education and age at first birth is almost 0.7 which is high. It is, therefore, difficult to be sure that the findings presented are attributable to educational polygenic score per se and not, for example, to age at first birth polygenic index. In the context of the paper, high genetic correlations between education and fertility-related outcomes are important because we know that parenting styles depend on fertility intentions, age, etc., and education polygenic scores might be capturing those as well. Therefore, the paper will benefit from additional reflections addressing this point.

Author Rebuttal, first revision:**Reviewer #1:**

Thank you for the response letter and the revised manuscript with tracked changes (minor thing: page numbers in the manuscript would have been helpful). I still like the work and the manuscript, but in general, I feel that the authors have mostly responded to the reviewers' comments in the letter but haven't done so much with incorporating our concerns in revising the manuscript. In particular, I'd like to see the following concerns better incorporated into the manuscript.

We thank the reviewer for these comments. We have made many more changes in this revised manuscript, as outlined below in response to each comment, whilst also being considerate of word limits. We have also inserted page numbers in the manuscript.

(1) I think the preprint by Nivard et al. is very relevant for the current manuscript and discussion on whether parental EA-genetics indicate the importance of parenting. It feels like the authors think so too, given the full-page discussion of this work in the response letter. It seems logical to me to reflect on this work in relation to their work in the manuscript. Yes, because of the heritability of parenting phenotypes, siblings will resemble each other in how they parent their offspring. This indeed fits with Nivard et al's finding that the average sib-ship's PGS/PGI in the parent generation explains variance in the offspring's

educational outcomes. But in the following analysis, Nivard et al find “find no evidence that parents’ PGI are specifically related to offspring academic achievement over and above the average PGI of the siblings in the parental generation.”

We agree with the reviewer, and we have now added our reflections to the discussion (pages 13/14). The new section reads: “Findings of genetic nurture have typically been interpreted as indicating an effect of parenting (or other parental characteristics) on offspring outcomes, but emerging research suggests a more nuanced interpretation. One recent study (Nivard et al., 2022; non-peer-reviewed preprint) found that parent’s education-associated genetics no longer predicted offspring educational outcomes over and above child genetics once parent’s siblings’ genetics were accounted for. This suggests that the processes mediating indirect genetic effects are not specific to one nuclear family, but shared across extended families. A challenge in comparing this and our study is that there are important differences in data and design: the study by Nivard et al. (2022) does not analyse parenting directly, whereas ours does; Nivard et al. uses a sibling design, whereas ours does not (due to lack of twins-as-parents data); Nivard et al. uses data from Norway, whereas ours uses data from the UK, New Zealand and the US; Nivard et al. specifically focuses on genetic nurture in relation to offspring educational attainment, whereas we analysed genetic associations with parenting behaviours that are linked to a broad range of offspring outcomes. Notwithstanding these differences, how do the findings compare? Our findings show some consistency with Nivard et al. (2022), because we observe genetic associations with parenting. Genetic associations with parenting imply that parenting will to some extent be shared across genetically-related family members. For example, because siblings share genes with each other, and genes are associated with parenting, siblings would naturally be expected to resemble each other in how they parent their offspring, more so than non-siblings. Likewise, there will be some expected parent-child similarity in parenting partly due to genes shared between parents and children. This suggests that parenting and parental investment are not exclusively within-family processes but are to some extent shared across related family members (including siblings and grandparents). This would lead to a reduction of genetic nurture within families, as seen in the study by Nivard et al. (2022) without necessarily negating a role for parenting in genetic nurture. However, if genetic nurture would be entirely attributable to parent behaviour, as it has been typically interpreted, one would expect to see at least some residual effect even after introducing a sibling control. Thus, the finding that the genetic nurture effect reduces to zero in the study by Nivard et al. (2022), if replicated, will raise important questions about the interpretation of genetic nurture, including to what extent it reflects processes such as population stratification. Neither the Nivard study nor our study can conclusively answer these questions. Addressing them will require an expansion of extended family studies (such as twins-as-parents studies) that collect molecular-genetic data alongside data on parenting and offspring outcomes.”

(2) As I said previously, the ultimate outcome would be the effect of parenting (and parental genetics) on offspring outcomes. In response to my comment, the authors have only added less than half of a sentence towards the end of the manuscript (“although our study does not look at child outcomes”). That doesn’t solve the problem of not being clear and open about what in the causal chain is and is not studied in the current manuscript. I’d suggest that the authors open up, perhaps illustrated in a figure in the introduction (a DAG?), about which part(s) of the causal chain between parenting and child outcomes are tested here and which are assumed based on the literature. Regarding the discussion of that literature, it helps if it’s clear to the reader what in the literature (on associations or effects of parenting on child outcomes) is merely correlational and what has been demonstrated as causal (or somewhere in between). Not all paths can be tested in one study, but it helps the reader to follow the current work and how that fits in the bigger picture.

We have added a new Figure to the paper (Figure 1, also added to this letter, see below), which depicts the paths we test in our study (paths a, b, and c) and those that we assume based on previous literature (paths d and e). We now walk readers through this Figure in the Introduction (page 5). Adding this Figure also means that we now state upfront (in the Introduction, rather than in the Discussion) that we did not analyse offspring outcomes.

We have also added a more extensive discussion of the literature that examines the assumed parts of the model, particularly path d from parenting to child outcomes. This is included in Supplementary Table 1 and provides an overview, for each developmental period, of the literature testing associations between parenting and child outcomes, and to what extent it is based on observational studies versus experimental and quasi-experimental studies.

Finally, although we agree with the reviewer that examining parenting is mainly important because of the assumed impact it has on child development (i.e., as the reviewer says, child outcomes are the ultimate outcome), we suggest that there is also value in analysing genetic and environmental influences on parenting for its own sake. This is because parental caregiving is such a key social behaviour; shows wide variation across individuals; and is an important part of life for a large part of the population (e.g., mothers in the UK spend on average 2 ½ hours each day with their children; fathers 1.5 hours: <https://ourworldindata.org/parents-time-with-kids>). Furthermore, associations between genetics and parenting are an excellent example of gene-environment correlation and “niche-building” whereby individuals construct and create an environment in line with their genetic proclivities. These processes are often studied in children, but much less in adults (and particularly, parents).

(3) I feel Reviewer's 2 concerns too are discussed at length in the response letter, but not so much incorporated in the manuscript.

Reviewer 2 made 7 comments that included questions about the results, or suggestions for changes to the manuscript. In our previous revision, we made changes in response to 5 of those comments, including changing the title of the manuscript; adding references to prior literature; clarifying text; and adding two paragraphs to the Discussion. In this revised version of the manuscript, we have additionally incorporated more of our response to two of reviewer 2's comments, which articulated that the results were not particularly surprising, especially given prior findings of phenotypic associations between education and parenting. We previously only discussed this point in our letter, but we have now inserted some of this discussion into the manuscript (pages 15/16): "The association between education-associated genetics and parenting may have been anticipated based on prior research findings of phenotypic associations between education and parenting. However, there is still value in testing these associations using genetic data. For one, a phenotypic association between education and parenting does not in itself mean that there must also be a genetic association between the two, especially given the fact that the polygenic score only explains a fraction of the heritability of educational attainment. Furthermore, the implications of a gene-environment correlation go beyond those of a phenotypic correlation between education and parenting, because a gene-environment correlation implies that a) genes may have environmentally-mediated effects via parenting; b) the unique contributions of genes and parenting cannot be easily separated in research that does not take both into account; and c) the advantage of inheriting genes linked to educational attainment is associated with the advantage of invested parenting, resulting in a 'double whammy' of education-associated genes and environments. Although previous studies have reported links between education-associated genes and some aspects of parenting during children's early years, here we show that these links are not unique to a few aspects of parenting in the early years of life, but are widespread across measures of parental investment, and that they continue and accumulate into offspring adulthood."

Reviewer #2:

This paper is a good addition to the literature about genetic influences on environmental measures ("nature of nurture"). Although "genetic nurture" is a well-established phenomenon (Kong et al., 2018, Willoughby et al., 2021), this study is valuable for its large sample size and the wide array of parenting traits it explores. I also liked the idea of controlling for offspring PGS as a way to approximate evocative rGE.

We thank the reviewer for their comments.

1) I think the main weakness of the study is that it only looks at maternal PGS, while examining paternal influences only as a sort of sensitivity analysis. For studies where both mothers and fathers are available, wouldn't it make more sense to use mid-parent PGS instead of only maternal PGS? This would also address concerns about the role of assortative mating. Parenting is usually a two-person job, and examining only maternal influence may tell only half the story.

We have added new analyses, using the subset of families in the MCS cohort where genetic data are available on mother, father and child, to incorporate fathers' polygenic scores. The findings show that fathers' polygenic scores were uniquely associated with several parenting outcomes, over and above mothers' and children's polygenic scores, specifically cognitive stimulation; warm, sensitive parenting; and health-parenting. This finding is very consistent with the reviewer's point that "parenting is a two-person job". It is also notable that in these models that included both parents' polygenic scores, children's polygenic scores were no longer uniquely associated with most parenting outcomes (except parental monitoring). This suggests that much of the apparent child effect on parenting that we saw in unadjusted models, or models that only include mothers' and child polygenic scores, may in fact reflect father's genetics. Furthermore, as predicted by the reviewer, we see evidence of assortative mating (mothers' and fathers' education polygenic scores were correlated $r=.14$). However, mothers' polygenic scores remained associated with most parenting outcomes after adjusting for fathers' polygenic scores, i.e. even after taking assortative mating into account. We now summarise these analyses in the manuscript (page 10) and report all results in Supplementary Figure 2.

2) As described by Table 1, childhood periods are covered by NZ and UK cohorts, while adulthood is covered by US cohorts. This is a potential source of bias, is it clearly acknowledged in the paper?

Thank you for pointing out this limitation, which is now clearly acknowledged in the paper (page 12: "Eighth, childhood periods were covered by NZ and UK cohorts, while adulthood is covered by US cohorts, which is a potential source of bias. This possibility can be tested once genetic data become available for more datasets (e.g. British cohorts of older adults)."

3) The authors report that "mothers with higher education polygenic scores were also less likely to drink heavily during pregnancy". It looks like this association is only significant in ALSPAC but not in MCS, so this should be acknowledged.

This is correct. We mentioned this previously, but have now made this much clearer in the text (page 6: “In the ALSPAC cohort but not in MCS, mothers with higher education polygenic scores were also less likely to drink heavily during pregnancy”).

4) In the analysis of intergenerational supports, the model controls for “respondent’s age, sex, assets/net worth, number of children, labour force status, year/wave, and proximity”. While some of the controls make sense (age, sex, number of children, year/wave, maybe proximity), the rest, especially assets and labour force status are clearly potential mediators of the association. Controlling for mediators may underestimate the effect that we are interested in (see the classic Meehl 1971: “High school yearbooks: a reply to Schwarz”). This may be responsible for the null association with wealth inheritance in the WLS.

Thank you for this comment. We completely agree with the reviewer that some of these variables may be mediators of the association. The reason why we reported results adjusted for assets in the Results was because we anticipated that readers might think that net worth/assets may explain observed associations between the genetics of educational attainment and parents’ inheritance behavior. In the revised manuscript, we now report the unadjusted estimates in the Results section, and add that adjusting for different covariates does not change the pattern of Results. Additionally, we now report more detailed results in the Supplement (Supplementary Table 2), which show that results do not change with the inclusion of different covariates.

Specifically, we estimated four models:

Model 1: Adjusted for wave/year, age, sex

Model 2: Adjusted for all the predictors as in Model 1, plus number of children (and, for childcare, proximity to children)

Model 3: Adjusted for all the predictors as in Model 2, plus labor force status

Model 4: Adjusted for all the predictors as in Model 2, plus assets/net worth

The results, in terms of statistical significance and interpretation, remain very similar across the four models.

5) The composite of parenting behaviours across time was only used in the E-Risk cohort, apparently because of its low attrition rate. Wouldn’t the power of the study be boosted by including all cohorts, perhaps using weights to account for differential attrition between them?

We have now repeated these analyses in the MCS and ALSPAC cohorts, and report the new results in the manuscript (pages 8/9 and Figure 5; the revised Figure 5 is included in this letter, see below). We find that the three key findings of the analysis replicate across cohorts: a) Parental investment was correlated across developmental periods, so that parents who invested more in one period did so in other periods as well; b) Mothers with higher polygenic scores tended to provide consistently greater parental investment across time; and c) The difference in polygenic score among mothers of children who received high parental investment in all versus none of the developmental periods amounted to approximately 0.8 standard deviations. We did also see some slight differences in results across cohorts, specifically: a) Although parental investment was correlated across developmental periods in all three cohorts, the magnitude of the correlations was highest in E-Risk and lower in MCS and ALSPAC, particularly for developmental periods further apart (e.g. between parental behaviour during the prenatal/infancy period, and adolescence); b) Although polygenic scores were associated with the consistency of parental investment in all three cohorts, the estimate in ALSPAC was a little lower than in E-Risk and MCS (E-Risk: $\beta=.23$ [95%CI .16, .29], $p<.01$; MCS: $\beta=.21$ [95%CI .18, .25], $p<.01$; ALSPAC: $\beta=.15$ [95%CI .11, .18], $p<.01$), and c) Although the difference in polygenic score among mothers who provided high parental investment in all versus none of the developmental periods was approximately 0.8 standard deviations across all three cohorts, the distribution of scores across these categories differed slightly across cohorts (as visible in Figure 5).

6) Two minor observations: 1) in line 357, there is an in-text reference. 2) in line 431 the authors refer to “the most recent GWAS of educational attainment” – this had been corrected in the introduction, it needs to be corrected there as well.

Thank you for spotting these typos, we have corrected them in this revised version (changed “the most recent” to “a recent” GWAS on page 17; and removed the in-text reference on page 13).

Reviewer #3:

This is a very well-written paper that was a pleasure to read. It is also very well-structured and straightforward. I also think it is an important paper for the field which links genetics and research on parenting. Reviewers from the previous round raised crucial questions, and I see the authors addressed these sufficiently and convincingly. I do not have much to add to those – the manuscript has been improved making it hard to criticise.

We thank the reviewer for these comments.

(1) My only question is about the polygenic score for educational attainment. While authors use it as the main analytic tool (which is meaningful), we know that this polygenic score has high genetic correlations with other outcomes. For example, as Mills et al. 2021 show the genetic correlation between education and age at first birth is almost 0.7 which is high. It is, therefore, difficult to be sure that the findings presented are attributable to educational polygenic score per se and not, for example, to age at first birth polygenic index. In the context of the paper, high genetic correlations between education and fertility-related outcomes are important because we know that parenting styles depend on fertility intentions, age, etc., and education polygenic scores might be capturing those as well. Therefore, the paper will benefit from additional reflections addressing this point.

We thank the reviewer for this comment and for the opportunity to reflect on this in the manuscript. We have added the following section to the Discussion (page 15): “Another possible explanation for why the education polygenic score is associated with parenting is through its genetic and phenotypic correlations with outcomes other than education, including fertility-related outcomes (such as age at first birth or number of children)⁴⁴ or mental health outcomes (e.g. depression)⁴⁵, which are all associated with parenting. Indeed, previous research in one of our cohorts, the Dunedin study, shows that the education polygenic score was associated with age at first birth, and that age at first birth was associated with parenting behaviour.¹⁴ However, the study also showed that age at first birth did not explain associations between education polygenic score and parenting behaviour.¹⁴ Consistent with this finding, in the current study, differences in parental investment observed in WLS and HRS were not explained by number of children. However, more research is needed to systematically compare associations between different polygenic scores and parenting.”

New Figure 1. A model of the associations tested or assumed in the present study.

Note to Figure 1. The Figure depicts the paths tested (in black) or assumed (in grey) in the present study. Path b is the path that we focus on in our study; it depicts the possibility that parents' genes are associated with the parenting they provide to their children. In order to test this possibility, it is necessary to control for path a, which depicts genetic transmission from (biological) parent to child; and path c, which depicts the possibility that children's genes are also associated with the parenting they receive (this is often referred to as evocative gene-environment correlation or child effects). If paths a and c are not controlled for, associations between parental genes and parenting may reflect genetic transmission and evocative gene-environment correlations (paths a*c). We therefore controlled for children's polygenic score in our models. Note that our study did not test offspring developmental outcomes (such as attainment or health outcomes), which are depicted at the bottom of Figure 1, and which are connected to the top of the Figure through paths d and e. It is assumed, based on previous literature (see Supplemental Table 1), that the genes parents pass on, and the parenting they provide, both impact on offspring outcomes (in the Figure, this is illustrated by paths a*d for genes, and paths b*e for parenting). Also note that even though the parent icon shows both mothers and fathers; most of our analyses used maternal polygenic score due to data availability; fathers' polygenic scores were analysed in a subset of models.

New Figure 5. Cumulation of parental investment across development in the E-Risk, MCS and ALSPAC cohorts.

Note to Figure 5: The Figure reports analyses from mothers and children participating in the E-Risk, MCS and ALSPAC cohorts. **Panels a, c and e** show tetrachoric correlations between parental investment across developmental periods. For each developmental period (prenatal, infancy, childhood, adolescence), a binary variable was constructed, capturing “high” parental investment (see Results for details). The colors in the matrices indicate strength of association, with darker areas indicating higher correlations. The panel shows that high parental investment in one period tended to be associated with high parental investment in other periods. **Panel b, d and f** report mean education polygenic scores for each category of a measure indicating the cumulation of parental investment across time. The measure was constructed by adding up the individual indicators of investment for each developmental period, so that the lowest scores indicated consistently low investment (“Always low”, n=189 in E-Risk; n= 425 in MCS; n=119 in ALSPAC) and the highest scores indicated consistently high investment (“Always high”, n=168 in E-Risk; n=255 in MCS; n=194 in ALSPAC). In between these extreme categories were categories indicating lower investment (n=460 in E-Risk; n=1,1313 in MCS; n=633 in ALSPAC), a moderate amount of investment (n=490 in E-Risk; n=2,594 in MCS; n=1,422 in ALSPAC) or higher investment (n=349 in E-Risk; n=1,655 in MCS; 1,123 in ALSPAC). The panel shows that mothers of children who received more consistently high investment across development tended to have higher polygenic scores, on average. The error bars indicate 95% confidence intervals.

New Supplementary Table S1. Evidence from previous research for associations between parental investment and child outcomes and for associations between children's genes and child outcomes.

Evidence for associations between parental investment and child outcomes	
Developmental period	Description of evidence
Prenatal period	Many observational studies report associations between prenatal smoking and heavy drinking and various child outcomes, including physical health outcomes (such as birth weight, BMI, asthma), ^{49–51} behaviour ⁵² and cognition. ⁵³ Although most of these studies control for confounders, they may still suffer from residual confounding, including from genetic influences. Evidence from RCTs or natural experiments, including genetically-sensitive designs, suggests effects of prenatal smoking predominantly on birth weight. ^{54–56} Likewise, most of the evidence for links between prenatal heavy drinking and many adverse child outcomes comes from observational studies; ^{57,58} evidence from quasi-experimental studies suggests a potential causal role of prenatal alcohol exposure on cognitive outcomes, and weaker evidence for a role in low birthweight. ⁵⁹
Infancy	As with prenatal smoking and heavy drinking, most of the evidence linking breastfeeding to child outcomes comes from observational studies. These studies show associations with many child outcomes, particularly childhood physical health outcomes such as obesity ⁶⁰ and asthma, ⁶¹ and with child cognitive outcomes. ⁶² As with prenatal smoking and heavy drinking, a threat to the interpretation of these results is that observational studies may suffer from residual confounding. A review of evidence from different study designs, including experimental and quasi-experimental studies, suggests effects of breastfeeding on cognitive ability. ⁶³
Childhood	A wealth of observational evidence reports associations between various dimensions of parenting and child outcomes. We focused on dimensions of parenting that have been most commonly examined in these studies and that have been most consistently associated with a wide variety of outcomes; these parenting dimensions include cognitive stimulation, ^{64,65} warm-sensitive parenting, ^{66–68} household chaos, ^{69,70} health-parenting (i.e. parent efforts at instilling healthy habits in their children e.g. via limiting screen time or providing healthy foods), ^{71,72} and support with schooling. ⁷³ These observational studies suffer from the same limitations as explained above,

	particularly the risk of residual confounding. However, there is some evidence from experimental and quasi-experimental designs to suggest a potential causal impact of these parenting dimensions for child outcomes, including evidence for effects of cognitive stimulation on child language outcomes,^{74,75} warm-sensitive parenting on externalising problems,⁷⁶⁻⁷⁸ household-chaos on externalising problems,⁷⁹ health-parenting on some child health outcomes,^{80,81} and school support on academic achievement.^{82,83}
Adolescence	One of the most well-researched aspects of parenting during adolescence is parental monitoring; numerous observational studies report associations between monitoring and offspring outcomes, particularly antisocial behaviour,⁶⁷ substance use and risky sexual behaviour,⁸⁴ and academic achievement.⁶⁸ Evidence from (quasi-)experimental research is more sparse, but suggests that parenting interventions during adolescence can reduce adolescents' risky substance-use and sexual behaviour.^{85,86}
Offspring adulthood	We focus on three common sources of intergenerational supports from parents to adult offspring: financial support, wealth inheritance, and childcare support. Perhaps unsurprisingly, previous research suggests that financial support and wealth inheritance increase offspring wealth, at least in the short term.⁸⁷⁻⁸⁹ For the provision of childcare support to the children of adult offspring, there is evidence from survey studies suggesting that it affects the labor market participation of mothers, as well as parents' fertility decisions.⁹⁰⁻⁹²
Evidence for associations between children's genes and child outcomes	
	Decades of evidence from twin and adoption studies show genetic influences on various offspring outcomes, including physical health, mental health, behavioural and educational outcomes.⁹³ More recent evidence for genetic influences comes from genome-wide association studies (GWAS) that have identified associations between measured genetic variation and various outcomes.⁹⁴ Findings from GWAS studies may suffer from several sources of confounding, such as indirect genetic effects, assortative mating or population stratification.⁹⁵ However, evidence from analyses of siblings (which control for potent sources of confounding) suggest that even among siblings born to the same biological parents, genetic differences continue to be associated with outcomes (although the magnitude of effects tends to reduce).⁹⁶

New Supplementary Table S2. Associations between parental polygenic score and intergenerational supports to adult offspring across models with adjustment for different sets of variables.

	Health and Retirement Study (HRS)			
	Model 1	Model 2	Model 3	Model 4
	RR (95%CI)	RR (95%CI)	RR (95%CI)	RR (95%CI)
Financial support	1.12 [1.10; 1.14]	1.12 [1.10; 1.14]	1.11 [1.09; 1.13]	1.10 [1.08; 1.12]
Help with childcare	1.03 [1.01; 1.06]	1.04 [1.02; 1.07]	1.05 [1.02; 1.07]	1.04 [1.01; 1.06]
Inheritance	® (95%CI) 0.12 [0.11; 0.14]	® (95%CI) 0.12 [0.11; 0.13]	® (95%CI) 0.12 [0.11; 0.13]	® (95%CI) 0.11 [0.10; 0.12]
	Wisconsin Longitudinal Study (WLS)			
	Model 1	Model 2	Model 3	Model 4
	RR (95%CI)	RR (95%CI)	RR (95%CI)	RR (95%CI)
Financial support	1.06 [1.04; 1.09]	1.07 [1.04; 1.09]	1.07 [1.04; 1.10]	1.07 [1.04; 1.09]
Help with childcare	1.10 [1.05; 1.14]	1.11 [1.07; 1.15]	1.11 [1.07; 1.16]	1.11 [1.07; 1.15]
Inheritance	1.00 [0.98; 1.02]	1.00 [0.98; 1.02]	1.00 [0.98; 1.02]	1.00 [0.98; 1.02]

Note: RR=Relative Risk; ®=Standardized regression coefficient; CI=Confidence interval.

Model 1: Adjusted for wave/year, age, sex

Model 2: Adjusted for all the predictors as in Model 1, plus number of children (and, for childcare, proximity to children)

Model 3: Adjusted for all the predictors as in Model 2, plus labour force status

Model 4: Adjusted for all the predictors as in Model 2, plus assets/net worth

New Supplementary Figure S2. Associations between mothers', father's and child polygenic scores in the MCS cohort.

Note: The figure shows standardised estimates of associations between mother, father and child education polygenic scores and parenting, during childhood and adolescence, both for mother, father and child polygenic scores individually (in orange) as well as the unique association for each score when in a model containing adjusting for the others (in blue). All analyses were done in the subset of MCS participants

who had genetic data and parenting data ($n=2,503$; with slightly lower n 's across parenting measures). Error bars indicate 95% confidence intervals.

Decision Letter, second revision:

17th January 2023

Dear Dr. Wertz,

Thank you for submitting your revised manuscript "Genetic associations with parental investment from conception to wealth inheritance in six cohorts" (NATHUMBEHAV-22020384B). It has now been seen by the original referees and their comments are below. As you can see, the reviewers find that the paper has improved in revision. We will therefore be happy in principle to publish it in Nature Human Behaviour, pending minor revisions to satisfy the referees' final requests and to comply with our editorial and formatting guidelines.

We are now performing detailed checks on your paper and will send you a checklist detailing our editorial and formatting requirements within a week. Please do not upload the final materials and make any revisions until you receive this additional information from us.

Sincerely,

Charlotte Payne

Charlotte Payne, PhD
Senior Editor
Nature Human Behaviour

Reviewer #3 (Remarks to the Author):

The authors have very successfully addressed my comments. Congratulations on the good work.

Reviewer #4 (Remarks to the Author):

I find authors' reflection is sufficient and it is well-placed in the discussion section.

Final Decision Letter:

Dear Dr Wertz,

We are pleased to inform you that your Article "Genetic associations with parental investment from conception to wealth inheritance in six cohorts", has now been accepted for publication in *Nature Human Behaviour*.

Please note that *Nature Human Behaviour* is a Transformative Journal (TJ). Authors whose manuscript was submitted on or after January 1st, 2021, may publish their research with us through the traditional subscription access route or make their paper immediately open access through payment of an article-processing charge (APC). Authors will not be required to make a final decision about access to their article until it has been accepted. IMPORTANT NOTE: Articles submitted before January 1st, 2021, are not eligible for Open Access publication. Find out more about Transformative Journals

With best regards,

Charlotte Payne

Charlotte Payne, PhD
Senior Editor
Nature Human Behaviour